# Multiplexed detection of SARS-CoV-2 and other respiratory infections in high throughput by SARSeq

Ramesh Yelagandula [1], Aleksandr Bykov [2], Alexander Vogt[3], Robert Heinen[1,2,4], Ezgi Özkan[2], Marcus Martin Strobl[1], Juliane Christina Baar [1], Kristina Uzunova[1,2,4], Bence Hajdusits[2], Darja Kordic[2], Erna Suljic[5], Amina Kurtovic-Kozaric[5], Sebija Izetbegovic[5], Justine Schaeffer[6,7], Peter Hufnagl[6], Alexander Zoufaly [8,9], Tamara Seitz[8], VCDI*, Manuela Födinger[9,10], Franz Allerberger [6], Alexander Stark [2,11], Luisa Cochella [2✉] & Ulrich Elling [1✉]

The COVID-19 pandemic has demonstrated the need for massively-parallel, cost-effective tests monitoring viral spread. Here we present SARSeq, *saliva analysis by RNA sequencing*, a method to detect SARS-CoV-2 and other respiratory viruses on tens of thousands of samples in parallel. SARSeq relies on next generation sequencing of multiple amplicons generated in a multiplexed RT-PCR reaction. Two-dimensional, unique dual indexing, using four indices per sample, enables unambiguous and scalable assignment of reads to individual samples. We calibrate SARSeq on SARS-CoV-2 synthetic RNA, virions, and hundreds of human samples of various types. Robustness and sensitivity were virtually identical to quantitative RT-PCR. Double-blinded benchmarking to gold standard quantitative-RT-PCR performed by human diagnostics laboratories confirms this high sensitivity. SARSeq can be used to detect Influenza A and B viruses and human rhinovirus in parallel, and can be expanded for detection of other pathogens. Thus, SARSeq is ideally suited for differential diagnostic of infections during a pandemic.

[1] Institute of Molecular Biotechnology of the Austrian Academy of Science (IMBA), Vienna BioCenter (VBC), Vienna, Austria. [2] Research Institute of Molecular Pathology (IMP), Vienna BioCenter (VBC), Vienna, Austria. [3] Vienna Biocenter Core Facilities GmbH (VBCF), Vienna, Austria. [4] Gregor Mendel Institute (GMI), Vienna BioCenter (VBC), Vienna, Austria. [5] Clinical Center of the University of Sarajevo, Sarajevo, Bosnia and Herzegovina. [6] Österreichische Agentur für Gesundheit und Ernährungssicherheit (AGES), Vienna, Austria. [7] EUPHEM Fellowship, European Centre for Disease Prevention and Control (ECDC), Stockholm, Sweden. [8] Department for Infectious Diseases, Clinic Favoriten, Vienna, Austria. [9] Faculty of Medicine, Sigmund Freud Private University, Vienna, Austria. [10] Institute of Laboratory Diagnostics, Clinic Favoriten, Vienna, Austria. [11] Medical University of Vienna, Vienna BioCenter (VBC), Vienna, Austria. *A list of authors and their affiliations appears at the end of the paper. ✉email: cochella@imp.ac.at; ulrich.elling@imba.oeaw.ac.at

Within just a few months, the newly emerged coronavirus SARS-CoV-2 caused the global COVID-19 pandemic[1]. While the world awaits widespread vaccination and effective antiviral therapies, several measures can prevent spread of the virus. Social distancing and more strict "lockdown" strategies are effective in containment but have a major negative impact on human well-being[2,3]. Therefore, the limited and directed application of such measures is desirable. Molecular testing for the presence of the virus by contact tracing and widespread surveillance of asymptomatic individuals, in particular for system relevant institutions and vulnerable person groups, can identify infection clusters and provide the information needed for directed quarantine or other containment measures[4–6]. Such massive testing has shown tremendous impact on containment of the spread of SARS-CoV-2 in China, South Korea, Taiwan, Singapore, and Slovakia[7–11].

Several methods have been put forward for assessing infection status, which fall into two categories: the ones that detect viral proteins (so-called antigen tests) typically from swabs, and those that detect the presence of viral RNA from swabs, pharyngeal lavage (gargle), sputum, bronchoalveolar lavage, or saliva samples[12–17]. Antigen tests facilitate some aspects of the logistics of mass testing and have recently proven useful in a country-wide effort in Slovakia[18]. However, their detection limit is typically hundreds of thousands of molecules per assay, which is borderline sensitivity to detect infectious individuals (1–10 million viral particles per swab)[19,20]. Moreover, antigen test development is time consuming and therefore does not represent a first-line strategy for future epidemic or pandemic outbreaks. Tests for the viral RNA typically rely on the detection of characteristic fragments of the viral genome or transcripts by reverse transcription (RT) and quantitative polymerase chain reaction (qPCR) and thereby offer higher sensitivity than antigen tests. They can also be rapidly adapted to new targets, SARS-CoV-2-specific qPCR assays for example were available already in January 2020[21]. Given that PCR reactions can amplify unspecific fragments despite the use of specific primer pairs (incorrect amplicons), widely used qPCR tests for COVID-19 use fluorescently labeled probes that signal the presence and abundance of sequence-matching amplicons only. This typically means that one or a few (2–3) amplicons can be detected per reaction, and that specific light cyclers are needed that can perform both PCR and fluorescence measurements. The scalability of such a method is limited by cost and equipment availability—primarily light cyclers.

A more scalable and cost-effective alternative is to couple the same RT-PCR reaction to next-generation sequencing (NGS) as a means of high-throughput readout. NGS-based approaches detect amplicon identity by sequencing and computational analysis and therefore are not limited in the number of different amplicons they can detect in parallel: multiple different fragments (viral and cellular controls) can be amplified per reaction, as long as primer pairs are compatible. In addition to detecting multiple fragments in parallel, individual samples can be uniquely labeled with characteristic sequence-identifiers, i.e., indices, to allow for pooled sequencing and subsequent computational deconvolution, directly obtaining a sample-specific readout. The advantages of detecting multiple pathogen amplicons per sample and processing tens of thousands of samples in parallel mean that NGS-based protocols offer huge cost-saving potential and are thus highly attractive for large-scale testing.

NGS protocols are conceptually simple and indeed a few different protocols have been developed and some even FDA approved[22–27]. Each of these methods have different strengths, yet also suffer from one or several challenges that directly impact sensitivity, specificity at the amplicon and sample level, scalability, and/or costs. In this work, we describe SARSeq (saliva analysis by RNA sequencing), a robust high-throughput protocol that overcomes these challenges by optimization of the initial sample conditions, a two-step endpoint RT-PCR, NGS-compatible amplicons with mutually compatible sets of primers, and a barcoding strategy that achieves perfect sample-recall by redundant dual indexing while scaling to tens of thousands of samples by combinatorial indexing along two dimensions. We apply this protocol to samples with synthetic RNAs and various different patient samples and demonstrate that it extends to the simultaneous detection of SARS-CoV-2, influenza viruses, and human rhinoviruses (HRV) from the same sample in a single experiment. Overall, our pipeline can be efficiently combined with high-throughput sample collection in 96-well formats, robotics and NGS to detect SARS-CoV-2 and other respiratory pathogens in tens of thousands of samples per experiment with a turnaround time of about 1 day (Fig. 1A).

## Results

**Two-step RT-PCR allows specific detection of SARS-CoV-2 from crude respiratory samples by NGS.** The first step toward establishing a high-throughput SARS-CoV-2 test was to find sample preparation methods that would bypass the costly and time-consuming steps of swabbing by medical staff and RNA purification from patient samples, while being compatible with RT-PCR. A number of sample types have been effectively used to detect SARS-CoV-2, including swabs collected in viral transport medium (VTM) or other buffers, saliva, and gargle with Hanks' balanced salt solution (HBSS) or saline solutions[13–15,17]. Gargle samples are a less invasive collection type, do not require medical staff, and show similar sensitivity to swabs collected by medical staff[15]. They are preferred to pure saliva as samples become more uniform in viscosity and are thus easier to pipette, a prerequisite for automation (Fig. 1A). Such samples, however, pose the challenge that exposure of viral or cellular RNA for RT must occur under strict inhibition of the high load of RNAses present in saliva[28]. A number of methods have been reported to expose and simultaneously stabilize RNA in these samples, including heat inactivating at 95 °C[22], treating with proteinase K[17], and mixing with TCEP/EDTA[29,30] or with QuickExtract solution[31]. To compare these methods, we obtained gargle samples (in HBSS) from one negative and two SARS-CoV-2-positive individuals, and either purified RNA or treated the gargle according to the different protocols. We then assessed RNA exposure and stability by performing TaqMan RT-qPCR of a virus-specific amplicon (N1) (Fig. 1B). All the methods generated stable RNA while maintaining similar sensitivity to purified RNA under our reaction conditions. We also tested QuickExtract and TCEP/EDTA on swab samples in VTM, in experiments that are described below. In these tests, QuickExtract showed the least precipitation upon heating to 95 °C and was thus used for most experiments unless otherwise stated. Our results are consistent with several other reports that RNA purification is not required for efficient SARS-CoV-2 RNA detection, enabling high-throughput downstream applications.

The next aspect we evaluated was when to add the DNA indices that distinguish individual samples. These can in principle be incorporated during the RT[22,25,26] as well as during the PCR[32,33], as extensions of the primers used to reverse transcribe or amplify the desired amplicons, respectively (Fig. 1C). However, we found that having primers with the required extensions during the RT step resulted in a large fraction of non-specific PCR products, presumably because the low temperature of the RT reaction allows substantial non-specific priming (Fig. 1D). Given the large but limited sequence space on NGS flow cells, such lack of specificity means that many more reads would be needed per

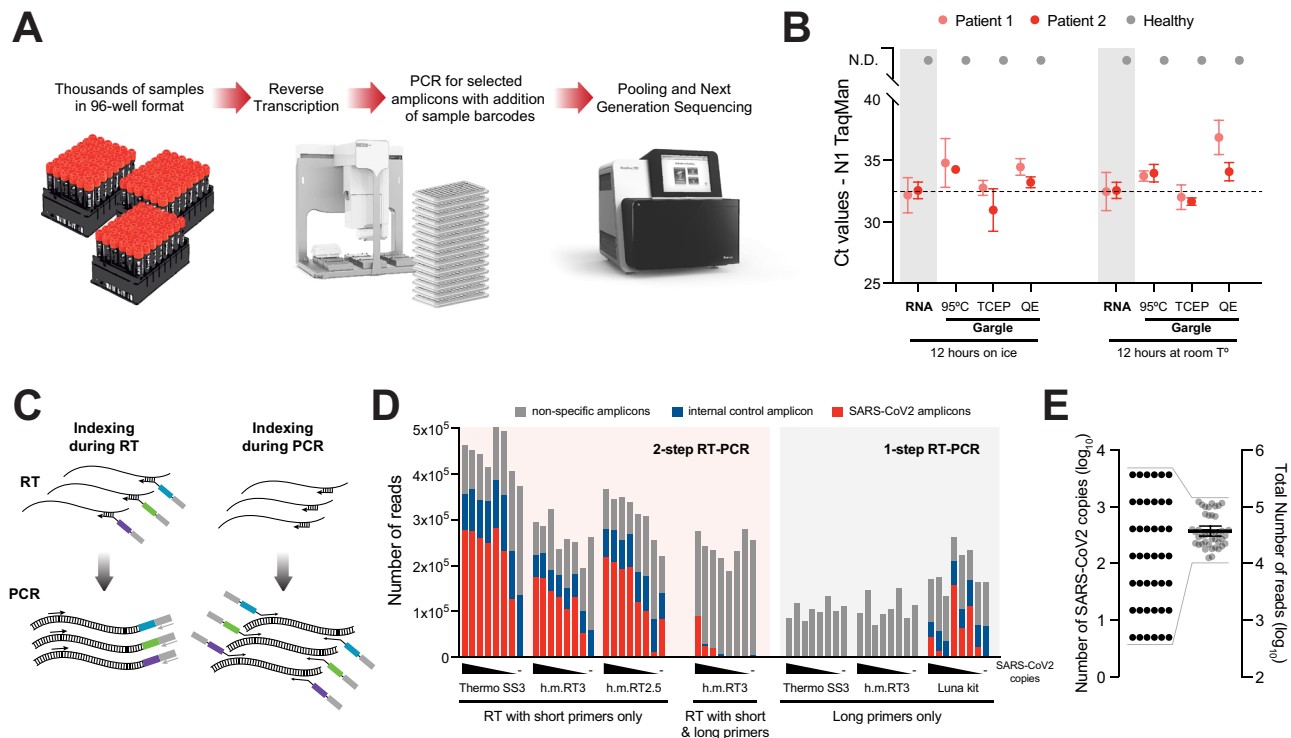

**Fig. 1 Two-step RT-PCR coupled to NGS allows specific detection of SARS-CoV-2 from crude respiratory samples. A** Overview scheme to illustrate the envisioned analysis pipeline. **B** Comparative analysis of SARS-CoV-2 detection by TaqMan RT-qPCR on the N1 amplicon, following different sample preparation treatments. Gargle samples from two SARS-CoV-2-positive and one negative person were collected in HBSS. RNA was purified or crude lysates were produced using either heat inactivation for 30 min[22], TCEP/EDTA (HUDSON buffer)[29,30], or QuickExtract[31]. Equivalent amounts of each treated samples were incubated overnight at room temperature or on ice to assess stability. Data are plotted as mean and SD of three replicates for each positive sample; $n = 1$ for each condition for the negative sample. **C** Illustration of amplicon indexing either during RT or PCR. Colored boxes symbolize different indices that label individual samples for identification by sequencing (by NGS). Arrows illustrate oligonucleotide hybridization to RNA during RT or DNA in PCR. **D** Comparison of amplicon levels measured by NGS after indexing either during RT or PCR. Amplicon-specific primers carrying extensions that contain the sample indices ("long primers") were either included as RT primers (indexing during RT) or used only in the PCR step (indexing during PCR); in the latter case, RT was performed with a mix of random hexamers and two N gene-specific primers ("short primers") (Supplementary Fig. 1). Indexing during RT was performed by either one-step RT-PCR (with Hot Start Taq polymerase already present during RT), or two-step RT-PCR (with Hot Start Taq polymerase added after RT). Black wedges symbolize a dilution series of synthetic SARS-CoV-2 RNA from 3645, 1215, 405, 135, 45, 5, to 0 molecules. Thermo SS3: Superscript III enzyme from Thermo Fisher; h.m. RT3: homemade Superscript III-like enzyme; h.m. RT2.5: homemade variant of Superscript II-like enzyme. **E** Total NGS read numbers per individual sample, for a set of 42 samples containing from 5 to 3645 SARS-CoV-2 RNA molecules; note the range compression from four to one order of magnitude, which enables equal representation of all samples across the sequencing space and therefore high sensitivity and scalability.

sample, limiting scalability to large sample numbers. We therefore chose a two-step reaction in which priming in the RT step is performed with random hexamers plus two gene-specific 12-mers that increase sensitivity for SARS-CoV-2 (Supplementary Fig. 1A, B), while integration of the sample indices occurs during the PCR. We also tested a number of RT and PCR conditions (Fig. 1D and Supplementary Fig. 1D, E) to arrive at the final conditions described in the Methods.

One of the hurdles toward establishing a pooled NGS-based assay for samples from virus-infected individuals derives from the fact that viral loads can differ by many orders of magnitude such that high-titer samples would dominate an NGS run. Diagnostic RT-qPCR for SARS-CoV-2 reports differences in Ct values of up to 25 cycles, which translate into $2^{25} = 33.5$ million-fold differences in viral titers. Therefore, if samples with low virus titer are to be robustly identified as positive, e.g., with >100 virus-derived amplicon reads, the samples with high virus titers would require $3.3 \times 10^9$ reads, which is prohibitive. For that reason, the dynamic range needs to be compressed to "dampen" the signals from highly positive samples while providing sufficient sensitivity to detect samples with lower titers. Of note, methods in which samples are indexed during RT (Fig. 1C) followed by PCR on

pooled cDNA samples would also maintain these quantitative differences, which is not desired for this application[26]. To achieve this compression, we ran the PCR reaction on individual samples for 45 cycles, until they reached saturation. This generated similar numbers of amplicons per well independent of initial viral titer (Fig. 1D, E and Supplementary Fig. 1C). In summary, using crude respiratory specimens as input, a two-step endpoint RT-PCR generates high specificity and uniform representation of correct amplicons across samples and enables pooling of many samples for analysis by NGS.

**Optimization of primer pairs targeting viral RNA and cellular control RNA.** In addition to the very large dynamic range of viral titers between patients, non-specific PCR amplicons can impair the detection of viral amplicons by NGS, because the number of NGS reads is inherently limited (and directly proportional to the total costs). For example, the parallel analysis of ~40,000 (96 × 384) samples means that each sample can receive a total of ~500 reads on a MiSeq, ~2000 reads on a HiSeq, and ~10,000 reads on a NextSeq platform. If a substantial fraction of these reads were spent on sequencing non-specific amplicons, assay sensitivity

would be severely impacted. It is thus pivotal to select amplicons and primer pairs that (i) show high sensitivity, (ii) generate amplicons of comparable short size, and (iii) generate few non-specific amplicons alone or in combination with any other primer present in the same reaction, which is of particular importance when using primers with long extensions (here: up to 42 nucleotides as PCR primers contain sample-identifying index sequences, staggers of random nucleotides, and primer binding sites for a second PCR as discussed below).

We thus tested several published SARS-CoV-2-specific primer pairs[21,34–36] after adding our index-containing extensions (Supplementary Fig. 2). We settled on the N gene-specific primer pairs N1 and N3 proposed by the Centers for Disease Control and Prevention (CDC) as they produce an ideal amplicon length of ~70 bp, performed very well in qPCR with detection by fluorescent stain (which does not control for amplicon identity) and showed good specificity in sequencing runs (Supplementary Fig. 2). Of note, other primer pairs with good specificity and sensitivity might be useful for parallel or backup detection strategies. We then tested the N1 amplicon together with the widely used internal control primer pair targeting *RPP30* (coding for RNAse P). While the N1 primers showed up to 50% of correct amplicons, in correlation with the amount of synthetically spiked in template, the fraction of specific amplicons for *RPP30* was only 0.06–1.5% (Supplementary Fig. 3A). When analyzing all sequenced amplicons across all samples shown in Supplementary Fig. 3A, we detected various short sequences that together made up >99% of all NGS reads. The vast majority was generated by the *RPP30*-specific primers (Supplementary Fig. 3B), suggesting that these primers are not compatible with our multiplexed PCR and NGS setup. We therefore set out to establish a new control primer pair that would produce fewer non-specific amplicons. We tested several primer pairs on gargle samples obtained from 16 individuals, yet only a single primer pair, specific for 18S ribosomal RNA, was detected in all samples and showed a strong dependency on the presence of reverse transcriptase (Supplementary Fig. 3C). Ribosomal amplicons were detected at Ct values of 15–45, a range comparable to that of viral amplicons in infected individuals. Upon further optimization of RT- and PCR-buffers (see Methods), we tested ribosomal RNA as a host control in combination with N1 primers. Under these conditions, the ribosomal primers generated a high fraction of specific reads both in the presence and absence of N gene template (Supplementary Fig. 3D).

Since the 18S amplicon—like the *RPP30* amplicon—does not span an intron, it cannot discriminate against genomic DNA templates abundant in respiratory samples. We thus designed an additional internal RNA control to assess successful RT in all samples independent of sample quality (RT control or reverse transcription control (RTC)); we produced an in vitro-transcribed RNA with identical primer binding sites as the ribosomal amplicon, yet an unrelated sequence in between (Supplementary Fig. 3E). This RTC was spiked into the RT mix at a concentration of 1000 molecules/reaction. Combining three primer pairs (N1, N3, and rRNA) and in the presence of the RTC, SARSeq reached high amplicon specificity with 30–80% amplicons corresponding to expected amplification products (Supplementary Fig. 3F and Fig. 1D). Moreover, the ratio between ribosomal amplicon and RTC reads provided a good assessment of sample quality: we observed that in the presence of good-quality gargle, the ratio is high and the RTC is lowly detected, but if the sample is low in nucleic acids the RTC takes over and the ratio is low. This strategy also monitors if samples contained RT and/or PCR inhibitors in which case neither amplicon is efficiently detected as we observed rarely in clinical samples (Supplementary Fig. 4A). The high specificity of amplicons achieved in our final setup (see

Supplementary Data 1 for pipetting schemes), and the even representation of reads across samples set the stage to develop a high-throughput indexing strategy that allows analysis of tens of thousands of samples in parallel.

**Two-dimensional redundant dual indexing allows scaling to population-level testing.** To exploit the high-throughput nature of NGS, we would need a sample barcoding strategy that allows multiplexing of tens of thousands of samples in a single sequencing run while retaining strict sample specificity, i.e., suppressing misassignment of reads to incorrect samples, which can lead to false-positive diagnoses. Several strategies for sample indexing are possible. First, samples can be individually indexed by a sample-specific short DNA sequence (typically called index or barcode) in the RT primer or one of the two PCR primers (Fig. 2A). In such a setup, sample-specific primers incorporate sample-indices into all amplicons from each sample, and these indices are then sequenced as part of the respective amplicon; e.g., Salis et al. designed 19,000 RT indices[26]. This strategy, however, does not scale well as it requires distinct primers for each amplicon and sample (linear/additive scaling). More importantly, it cannot retain perfect sample identity due to template-switching PCR artifacts[37,38] and index-hopping on flow cells[39,40], which can lead to incorrect associations between amplicon and index sequences. This problem is of particular relevance when high-titer samples are analyzed next to samples from healthy individuals as we demonstrated by spiking synthetic SARS-CoV-2 template into two wells of a 96-well plate (wells B8 and F2) in which all other wells are negative (Fig. 2B). The scalability limitation can be overcome with a combinatorial indexing strategy, such as a column index on the forward primer and a row index on the reverse primer[41] (combinatorial/multiplicative scaling; Fig. 2C). However, such a strategy suffers from the same inability to retain perfect sample identity, which in this case leads to a characteristic cross-shaped pattern along the rows and columns of the positive samples due to the misassignment of the row or column indices (Fig. 2D).

We developed an indexing scheme for SARSeq that achieves perfect sample specificity and combinatorial scalability. Specificity regarding sample identity was achieved by two indices that both point to the same sample/well (Fig. 2E, F), a strategy termed redundant dual indexing or unique dual indexing[41,42]. These two indices are introduced through forward and reverse primers and redundantly encode each sample with distinct indices at each end of the amplicon, thereby eliminating illegitimate index combinations. Such an approach requires two indices (=unique primers) per sample and therefore does not scale well when a single PCR is used (one dimension). We therefore use a two-dimensional indexing strategy, which we realized by two subsequent PCR steps: after the first PCR performed with unique dual indexing, we pool all samples within one plate into one well of a second plate and perform a second PCR that again uses unique dual indexing. This strategy of two-dimensional redundant dual indexing allows combinatorial indexing between dimension 1 and 2, and thus multiplicative scaling, while retaining perfect sample identity (Fig. 2G, H). It requires an only modestly higher number of indexing primers for very many samples and allows the encoding of $96 \times 96$ or $96 \times 384$ samples with $2 \times 96$ amplicon-specific primers (2 per amplicon; first dimension) plus $2 \times 96$ or $2 \times 384$ global primers (irrespective of amplicon; second dimension), respectively.

In practice, we extended the amplicon-specific primers for the first PCR (first dimension) at their 3′ ends to include a sample-specific index and i5/i7 sequences as primer binding sites for the second PCR. To ensure sufficiently complex sequences of the

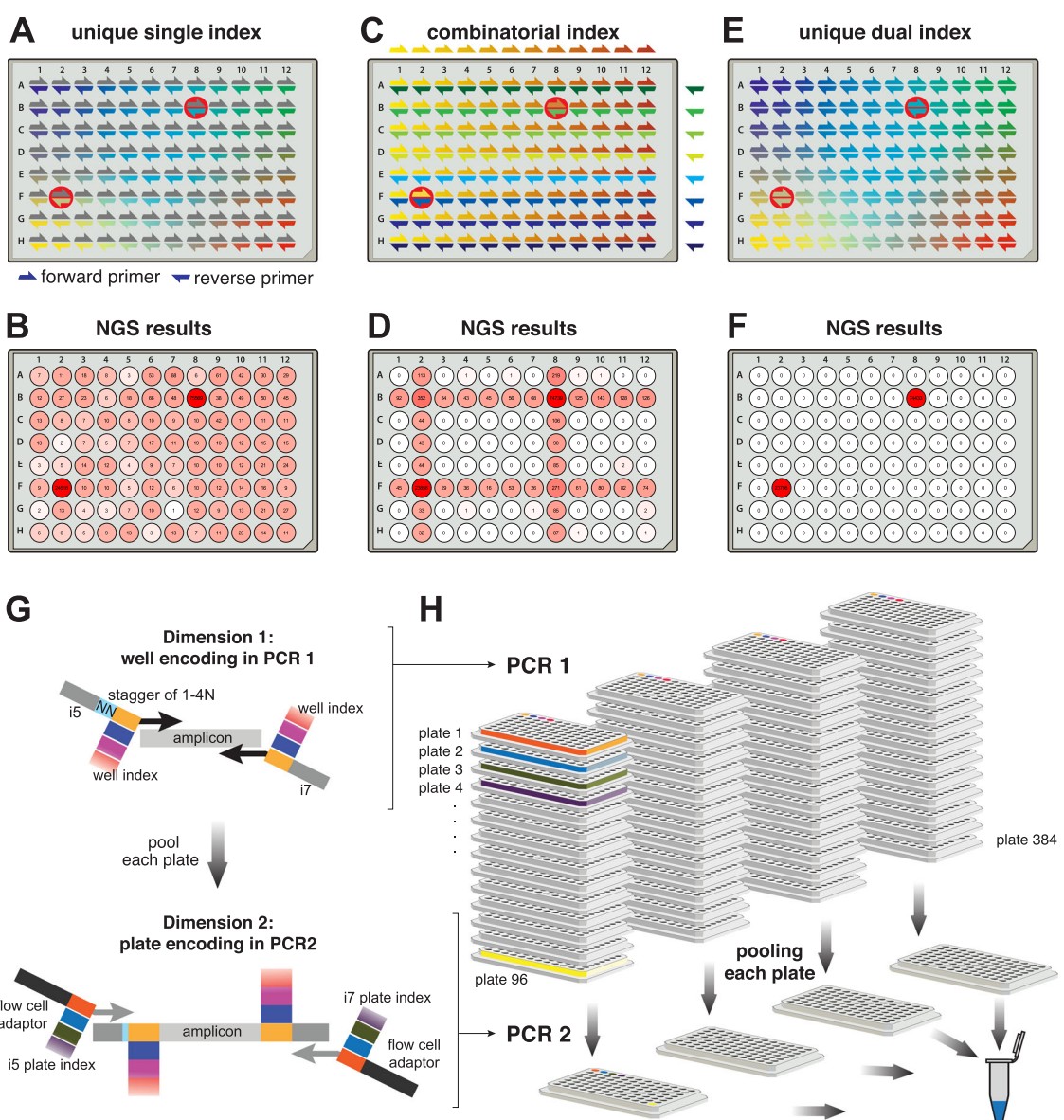

**Fig. 2 Two-dimensional redundant dual indexing allows scaling to population-level testing. A** Scheme depicting 96-well-specific indices. These can be incorporated by forward or reverse primers. For the latter they can be incorporated during RT or PCR. Red circles highlight the positions into which synthetic SARS-CoV-2 RNA was added to test specificity of the indexing strategy. **B** N3 amplicon reads obtained by NGS and mapped to each well based on indices incorporated by reverse primers during PCR. Forward primer indices were present but disregarded in this analysis. Note the frequent misassignment to incorrect positions. **C** Scheme depicting combinatorial indexing. Each well is identified as a unique combination of a forward and a reverse index. **D** N3 amplicon reads mapped to wells based on combinatorial indexing as in **C**. To simulate combinatorial indexing the identical dataset as in **B** and **F** was used but for analysis, primers were treated in pools pointing to columns or rows. **E** Unique dual indexing is a redundant indexing method that encodes each well both by a unique forward and a unique reverse index. Thus, illegitimate recombination products between an amplicon and its associated indices can be bioinformatically rejected. **F** NGS result for the same dataset as in **B** and **D** analyzed with unique dual indices successfully filtered away all misassigned reads. **G** Two rounds of unique dual indexing (in two subsequent PCR reactions) can be used to index first wells and then plates, effectively achieving combinatorial (multiplicative) scaling. Colored boxes represent indices. **H** Illustration of the PCR workflow. RT and PCR1 are performed on all samples individually, adding well-specific indices. Subsequently, each plate is pooled to one well of a new plate, reactions are treated with Exostar to remove excess primers and a second PCR is done, adding plate-specific indices and the Illumina flow cell adaptors. Our currently used and validated index set allows pooling of up to 36,864 samples (96 in PCR1 × 384 in PCR2). Finally, PCR2 amplicons are pooled, gel purified, and sequenced on any Illumina platform.

NGS forward reads for cluster identification and stable sequencing, we staggered the sample-index and the amplicon-specific sequence by a random offset of one to four base pairs (Supplementary Data 2). We tested a total of 110 primer pairs per amplicon (N1, N3, and 18S rRNA) to establish a set of 96-primer pairs that show good amplification behavior for all amplicons (Supplementary Fig. 4B). For the final set of primers, all amplicons within one well obtain identical offsets and indices to prevent recombination between amplicons and indices during PCR and to simplify bioinformatic analysis. Pre-prepared primer-plates and robotic pipetting pipelines allow us to process thousands of samples in parallel.

After the indexing of individual samples (=wells of a 96-well plate; first dimension), all samples of one plate were pooled to one

position of a new 96-well plate, and in a second PCR, a plate-specific index was added (second dimension). We implemented three measures to ensure that sample identity was perfectly retained between the first and second PCR. First, to ensure that primers from the first PCR were used up in PCR1 and thus not present during the second PCR, we included an RNA template with N1 and N3 primer binding sites similar to the RTC and the normalization-spike-in used in the SwabSeq pipeline[15] (Fig. 3A). Second, we treated the pools of PCR1 with DNA exonuclease to enzymatically degrade all single-stranded DNA and thus all remaining primers especially from SARS-CoV-2-negative wells. Third, we kept the cycle number for PCR2 at a minimum to avoid amplicon recombination during PCR and used a PCR protocol that prevents premature termination of an extension step[43]. Indeed, all three measures synergistically contributed to the robustness of read assignment (Supplementary Fig. 5A). In each dimension we used dual and redundant indices with a Hamming distance of at least three mismatches. The primers used for PCR2 (second dimension) are Nextera primer sets that are commercially available as 384 unique pairs and frequently used in many NGS sequencing facilities for multiplexing. This pipeline also suppressed read misassignment across plates, such that the highly positive positions displayed in Fig. 2A, C, E did not produce false-positive signals in any of the other, negative control plates within the same experiment (Supplementary Fig. 5B).

Our experimental design provides scalable and robust indexing for SARS-CoV-2 amplicons by using PCR to incorporate two sets of redundant indices, and bioinformatics to only allow legitimate combinations of those four indices. The ability to encode 96-well indices (first dimension) and 384-plate indices (second dimension) means this can be used to prepare 36,864 individual samples simultaneously. To illustrate scalability of the approach, a four-fold increase in one dimension for example by using 384-well indices in the first PCR would enable multiplexing of >145,000 samples. This degree of multiplexing renders the sequencing price per sample negligible and thus enables frequent population-wide testing because sequencing capacity is also not bottleneck with current NGS platforms. In summary, redundant dual indexing ensures sample identity specificity and two-dimensional indexing allows scalability while preventing any spill of reads from positive to negative samples even across multiple orders of magnitude in signal intensity.

**SARSeq is specific and sensitive when tested on a large set of gargle samples.** To test sensitivity, specificity, and scalability of SARSeq we set out to run large sample cohorts in which we diluted synthetic RNAs or a high-titer patient sample into SARS-CoV-2-negative gargle samples from hundreds of different people. In addition, we also used this setup to test the effect of spike-ins with identical primer binding sites to the N1/N3 amplicons, but different sequences in between, as introduced previously[15] (Fig. 3A). We processed multiple 96-well sample plates in parallel using a robotic pipetting platform.

We first assessed the sensitivity of SARSeq in this setup by diluting viral templates to 1, 3, or 10 copies per reaction (0.2–2 copies/$\mu$L in the 5 $\mu$L sample input). To account for the contribution of QuickExtract as well as to test RNA exposure from viral particles, we diluted synthetic RNA in $H_2O$ as well as QuickExtract:HBSS but also virions packaged in cell culture, in QuickExtract:HBSS. We measured each dilution in 24 replicates using SARSeq as described above. Using $H_2O$ for dilution we detected the 1 copy solution of SARS-CoV-2 RNA in 5 and 6 of 24 tested cases for N1 and N3, respectively. At such dilution, assuming a Poisson distribution, 63% of wells are expected to contain one or more viral copies, pointing toward a detection

efficiency of 30–40% per molecule. QuickExtract increased that efficiency to 70% while detection straight from viral particles was at 1.1 per molecule (Fig. 3B and Supplementary Fig. 6). COVID-19 patients have been classified as infectious with viral titers of >$10^6$/mL measured from 3 mL swabs[44]; our detection limit is thus at least 100 times more sensitive than required for mitigation strategies for the SARS-CoV-2 pandemic[45].

We had introduced the spike-ins containing the N1 and N3 priming sites to ensure that these primers are used up even in the absence of viral templates (Fig. 3A). We wondered if possible primer competition would thus decrease sensitivity of the assay. Satisfyingly, the presence of 100 copies of each the N1 and N3 spike-ins did not decrease sensitivity, at least in part because the viral amplicon outcompetes the longer spike-ins (Fig. 3B). Due to the endpoint PCR we perform, SARSeq intentionally blunts quantitative differences in the returned reads (Fig. 1E), yet a more quantitative readout can be desirable. Spike-ins have been used to improve the quantitative ability of other NGS approaches that rely on endpoint PCR (the ratio between viral amplicon and spike-in reads reflects the ratio of these two templates in the starting reaction[22]), we therefore tested the effect of different spike-in levels on the quantitative behavior of SARSeq. We also tested the effect of modulating PCR cycle number on sensitivity and quantitativeness (Supplementary Fig. 7). The number of spike in molecules was optimal at 1000/reaction (Supplementary Fig. 7 and Fig. 3C–E). We found the ratio of specific reads to spike-ins to serve as a good quantitative readout independent of the number of PCR cycles performed across the entire tested range of three orders of magnitude (Fig. 3D). If desired, sensitivity can also be lowered by reducing the number of PCR cycles below 45, e.g., at 37 cycles only viral titers of >200/$\mu$L ($10^3$/5 $\mu$L reaction input) are robustly detected.

To challenge SARSeq with hundreds of real samples while omitting RNA purification and thus prepare for a clinical performance study, we generated sample plates from pharyngeal lavage (gargle) collected in HBSS from healthy participants of routine SARS-CoV-2 testing at our institutes. Such diverse, crude samples may contain reagents inhibitory to the RT or PCR step. All gargle samples were previously tested negative through qPCR but we added synthetic SARS-CoV-2 RNA and a dilution series of a positive gargle sample (with Ct = 30 based on RT-qPCR with fluorescent probe) into several marked positions (9 out of 30 plates are shown in Fig. 3F). Subsequent to PCR1 we scaled up the experiment by creating six replicates of each sample plate and thus effectively generated 180 pooled plates that were processed in parallel in PCR2 and sequenced on one NextSeq high output lane. We therefore measured a total of 2880 real or 17,280 replicated samples in this batch. All samples returned high read counts for ribosome or RTC, excluding broad presence of RT or PCR inhibitors in these samples. We detected all positive samples with both N1 and N3 amplicons (with the anticipated exception of a $10^{-5}$ dilution of the positive gargle sample with a Ct value of 30), suggesting very high sensitivity (Fig. 3F).

Another critical parameter when testing large numbers of patients is the false-positive rate. We were therefore pleased to see that our indexing strategy and pipeline delivered typically zero and very rarely 1 read indicative of SARS-CoV-2 for gargle samples previously tested negative by qPCR as well as for all $H_2O$ controls, compared to hundreds or thousands of reads for positive samples. In total we performed four runs using gargle samples from our in-house testing pipeline, adding up to 4952 negative samples and 728 positive samples created by adding synthetic SARS-CoV-2 RNA or dilutions of a positive patient sample. We observed two unexpected N1-positive samples and five unexpected N3-positive samples, estimating a false-positive rate for our pipeline of 0.04–0.1%. This binary result showcased an

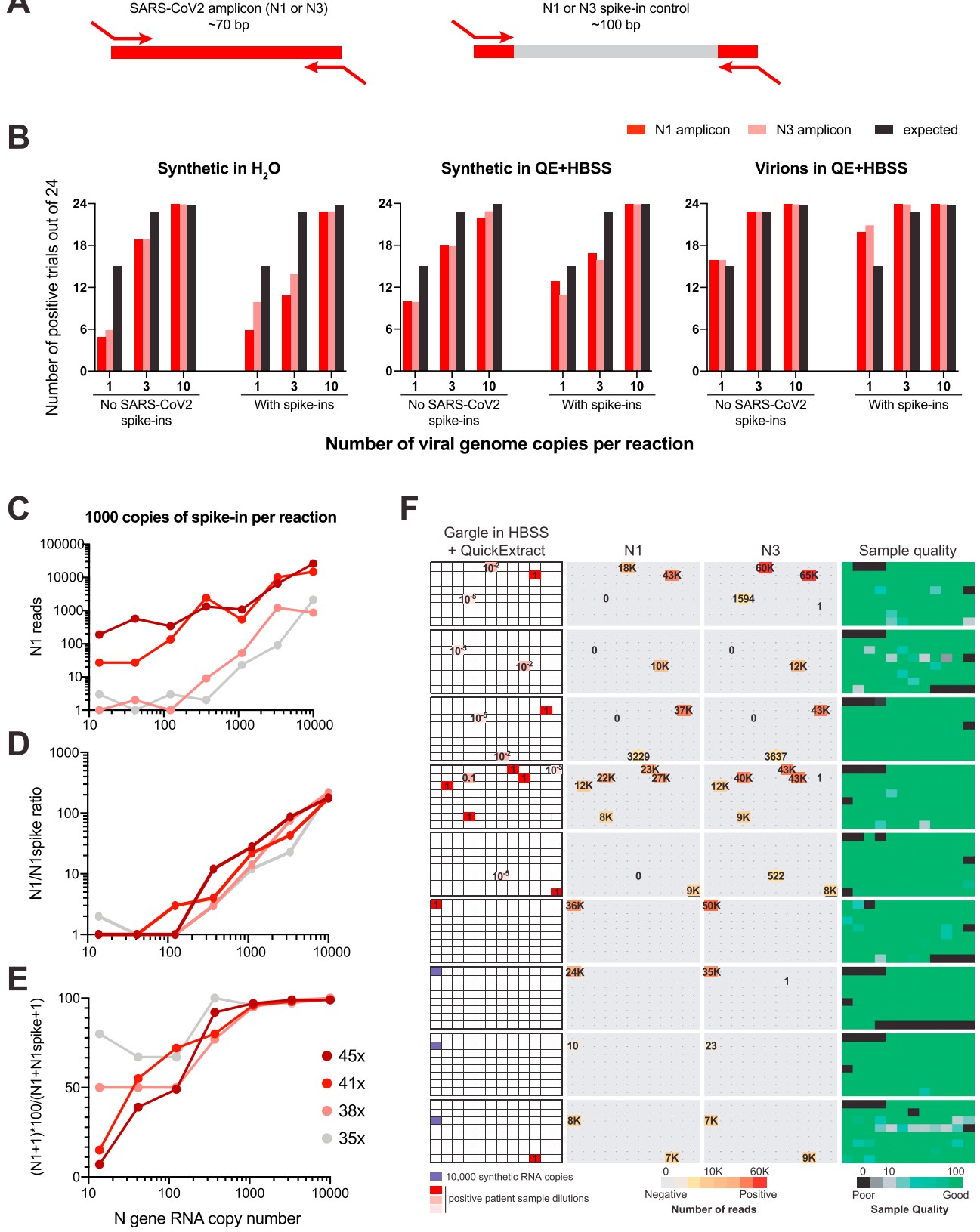

unambiguous assessment of infection status by SARSeq. Due to the absence of false-positive reads we also did not need to further use the N1 and N3 spike-in "denominator" amplicons to set a threshold ratio for calling positive results[22]. In summary, SARSeq enables the semiquantitative assessment of synthetic SARS-CoV-2 RNA in various buffers and in gargle samples and allows the

detection of SARS-CoV-2 RNA with high sensitivity and specificity.

**SARSeq robustly detects SARS-CoV-2 in patient samples from a clinical setting**. To test if SARSeq robustly detects real SARS-

**Fig. 3 SARSeq is specific and sensitive when tested on a large set of gargle samples. A** Schematic illustration of RNA spike-ins to scavenge N1 and N3 primers during PCR1 in SARS-CoV-2-negative samples. Analogous to the RTC, primer binding sites are identical to N1/3 amplicons, but intermediate sequence is distinct and longer so as not to compete with virus-derived amplicons. **B** Frequency of detection of SARS-CoV-2 at minimal template concentration. Number of detected cases out of 24 trials is depicted. Note that the expected frequency of detection is only 63% and 95% for one and three molecules, respectively, assuming a Poisson distribution of molecules in wells. **C** Read counts of N1 amplicons in a synthetic SARS-CoV-2 N gene RNA dilution series, in the presence of 1000 copies of N1 spike-in per reaction, obtained after 35, 38, 41, or 45 PCR1 cycles. **D** Ratio of N1 to N1 spike-in reads from the same experiment as in **C**. **E** Percentage of specific reads generated by the N1 primer pair that correspond to the N1 amplicon from the same experiment as in **C**. **F** SARSeq performance on a test pool of 864 gargle samples collected in HBSS from healthy participants of a routine SARS-CoV-2 testing pipeline. These were spiked with synthetic SARS-CoV-2 RNA or a dilution series generated from a positive patient sample (with a Ct = 30). All negative samples produced 0–1 N1/3 reads, while positive samples produced thousands. The only missed samples were $10^{-5}$-fold dilutions of the patient sample, which presumably did not contain SARS-CoV-2 RNA anymore at that concentration. Sample quality is assessed by the ratio or ribosomal reads to RTC spike-in.

CoV-2 virus in patient samples collected in clinical diagnostic settings, we measured a set of 564 swab samples from independent patients collected at the Clinical Center of the University of Sarajevo (Sarajevo, Bosnia and Herzegovina). These samples were obtained in VTM inactivated in QuickExtract and measured in duplicates. Both qPCR and SARSeq were performed from these crude lysates to compare sensitivities when using the same input material. While we did not detect N1 or N3 amplicons in $H_2O$ or RT conditions (Fig. 4A, B), we frequently obtained SARS-CoV-2 reads across all plates with good correspondence between amplicons and replicates (Fig. 4C, D). To assay correspondence to the standard test, we also measured the samples in a probe-based RT-qPCR assay in duplicates (also without prior RNA purification). As expected, we observed a robust correlation of both qPCR replicates until Ct ≈ 36 (Fig. 4E, red dots) and stochastic behavior beyond that detection limit (Fig. 4E, orange dots) with either one or two replicates scoring positive. No SARS-CoV-2 was detected in an additional 354 samples. We analyzed the overlap between detection by qPCR and NGS and found that 157 (96.3%) of samples with a Ct value of <36 in at least one qPCR scored positive in all four assays, while six samples scored positive in one or two assays only (Fig. 4F). Importantly, sensitivity of probe-based RT-qPCR and NGS on these samples was therefore fully on par. We hypothesized that the stochastic behavior at the detection border is due to the presence or absence of a single reverse-transcribed viral genome. If that was true, detecting that genome by both amplicons within one replicate should be more likely than detecting one of the two amplicons across both independent replicates. In contrast, different sensivities toward the N1 and N3 amplicons would result in the opposite outcome. Indeed, of the samples detected with one to three of these assays, 48 and 43 patient samples showed detection of SARS-CoV-2 with both amplicons within one SARSeq replicate, while only 18 and 25 samples were detected twice independently with N1 or N3 amplicon, respectively. We thus conclude that SARSeq appears sensitive down to single reverse-transcribed viral genomes and that the sensitivity is on par with qPCR if the sample preparation is identical. We also assessed the reproducibility of SARSeq runs (Supplementary Fig. 8A, B). As anticipated, the absolute read numbers are not necessarily correlated, but there is very good correspondence regarding whether or not a sample is positive. We observed the expected upper end of N1/N3 read counts produced by the endpoint PCR strategy to distribute sequencing space evenly across samples (Fig. 1E). In conclusion, SARSeq detected all samples that were reproducibly detected by qPCR. At the detection limit, both SARSeq and qPCR detected additional samples in a stochastic, non-reproducible manner as expected.

**SARSeq robustly detects SARS-CoV-2 in samples from a human diagnostic setting.** As a further pilot for a systematic clinical performance study we compared SARSeq to a gold

standard qPCR assay conducted in a human diagnostic setting. This test included 90 samples of diverse nature, including swabs in VTM, gargle in isotonic NaCl, and others, of which 28 were positive according to a gold standard qPCR pipeline including RNA purification. This pipeline used the equivalent of 17 µL crude sample as assay input whereas we used 2.5 µL (for Quick-Extract mix) and 4.7 µL (for TCEP/EDTA mix) crude original sample as input for SARSeq.

We performed seven replicates for SARSeq (three in Quick-Extract and four in TCEP/EDTA) that yielded highly consistent results: as expected, samples that were negative by qPCR showed only 0 or 1 reads in all replicates, whereas samples that were positive by qPCR consistently displayed thousands of reads. Specifically, we detected the N1 amplicon in 7/7 replicates for all positive samples with Ct values <36.5 and in at least 1/7 replicates for all others with Ct values <38.9 (Fig. 4G). The N3 amplicon showed a similar pattern but seems more sensitive to sample quality (Fig. 4H). We also assessed quantitativeness by comparing Ct values from the qPCR directly to the read counts obtained by NGS (Supplementary Fig. 8C, D). As intended by the endpoint PCR for dynamic-range compression, SARSeq is blunted for high viral titers (Ct values lower than ~33) but is semiquantitative for weakly positive samples. We therefore conclude that SARSeq is a useful method to detect SARS-CoV-2 in clinical and diagnostic settings for samples of various chemical compositions, robustly detecting samples with Ct values ~36, but also samples up to Ct values of 39, albeit with decreasing probabilities.

**SARSeq can detect multiple respiratory viruses in a single reaction.** Multiple infectious agents cause diseases with overlapping clinical symptoms to COVID-19, including influenza A and B virus, parainfluenza virus, rhinoviruses, and respiratory syncytial virus. It is expected that, particularly in the winter season, various respiratory symptoms will cause concerns and thereby dramatically increase the demand for SARS-CoV-2 tests. For SARSeq, adding amplicons corresponding to other infectious agents comes at little extra cost as long as it does not increase the required sequencing depth. Therefore, we can further multiplex SARSeq to detect other common respiratory viruses (or other pathogens) found in the same sample used for SARS-CoV-2 testing.

As proof of principle, we optimized primers for influenza A virus, influenza B virus, and rhinovirus to be combined with our SARS-CoV-2-specific SARSeq pipeline (Supplementary Fig. 9A). To this end we selected primers based on qPCR performance, amplicon length, and an NGS pilot experiment. For a pan-influenza A amplicon we settled on combining a degenerated forward primer from Bose et al. with a degenerate WHO reverse primer, both targeting the M gene[46,47]. For pan-influenza B, we selected a previously characterized primer pair binding to the M

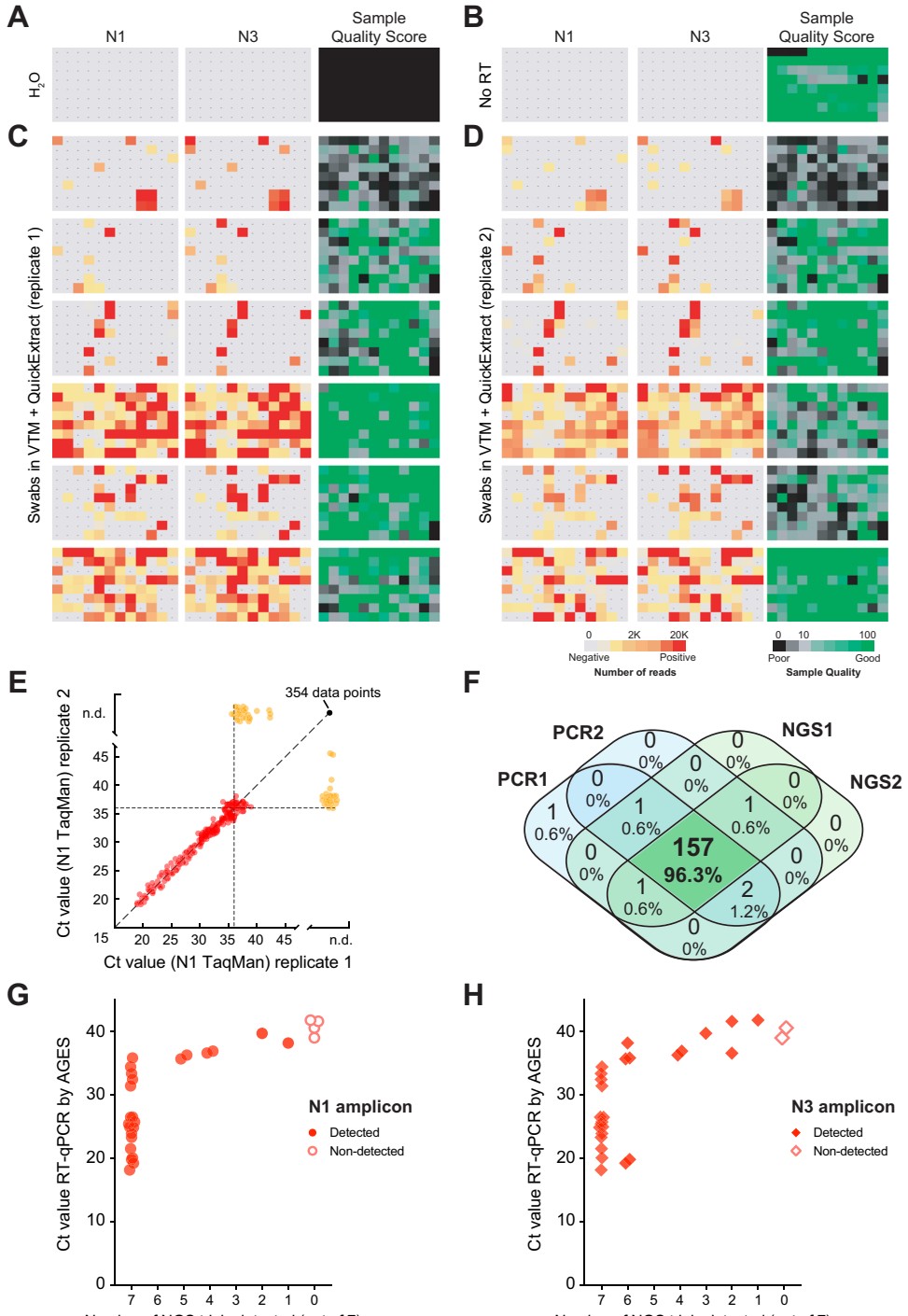

**Fig. 4 SARSeq robustly detects SARS-CoV-2 in patient samples. A** $H_2O$ control plate of SARSeq with three primer pairs, namely N1, N3, and ribosome in the presence of RTC as well as N1 and N3 scavenger spike-ins. **B** SARSeq on a gargle-QuickExtract plate in the absence of reverse transcriptase. No false-positive wells were detected. Sample QC scores high due to the absence of RTC reads while DNA templated ribosomal amplicon is observed. **C** Analysis of 576 samples obtained by swab and collected in VTM. **D** Independent replicate of **C** based on the same inactivated patient swab. Color codes depict read counts and sample quality score for **C** and **D**. **E** Two independent N1 TaqMan qPCR runs on samples used for **C** and **D**. Ct values of both runs are plotted against each other. Color code: red: sample scoring positive in both qPCR replicates, orange: sample scoring once, black: sample scoring negative. Stochastic and latest detection of SARS-CoV-2 at cycle 36. **F** Venn diagram of reproducibility for all samples scoring positive with Ct 36 or less for at least one qPCR replicate. **G**, **H** Comparison of NGS results by SARSeq to Ct values obtained by diagnostic qPCR. A set of 90 samples (including swabs and gargle in different buffers) was used for RNA extraction and qPCR measurements and in parallel aliquots of these samples were mixed either with QuickExtract or TCEP/EDTA and measured by SARSeq in triplicates and quadruplicates, respectively. We report the number of replicates in which we called a sample positive by NGS (with the N1 or N3 amplicon) relative to the qPCR Ct values. Not shown are 63 samples that were negative by qPCR and NGS.

gene[47]. Rhinovirus was detected using a primer pair described previously[48].

To test performance across a large number of specimens, we used sample plates from gargle collected in HBSS via our in-house testing pipeline (negative for SARS-CoV-2) and spiked in purified RNA obtained from HEK293T cells infected with respective virus strains at a ratio of 1:100 (per gargle volume) or dilutions thereof. Samples were processed using the protocol and robotic pipeline as for other experiments, except that we performed PCR1 in the presence of additional primer pairs, two against SARS-CoV-2 (N1 and N3), and one each for ribosomal control, influenza A, and rhinovirus. We initially tested the setup with 12 sample indices, so upon pooling sets of 12 samples and subsequent PCR 2, samples were sequenced and reads were mapped back to individual wells (Fig. 5). Given the observed robust performance, we expanded the set of primers specific for Influenza A and B viruses to the full set of 96-sample indices, and performed another spike-in test based on negative clinical samples collected in VTM and inactivated in QuickExtract (Supplementary Fig. 9B). Whereas this requires further sensitivity analyses on influenza-positive patient samples, SARSeq provides a highly specific and robust assay for detection of multiple viruses across tens of thousands of samples in parallel.

In summary, our multiplexed pipeline to detect RNA of various viral respiratory diseases in parallel performed robustly across multiple 96-well plates. These multiple virus experiments did not produce any false negatives with the exception, as expected, of the 1/1mio dilutions of a SARS-CoV-2 sample with Ct = 30. We also did not see any false positives. Generally, false positives in our assay can only arise from cross-contamination, so to avoid this all post-PCR steps were always performed in a different lab from the initial setup of the reaction, preventing contamination by DNA amplicons. In addition, we implemented incorporation of UTP and a UDG digestion step prior to PCR1 in our protocol, further reducing the risk of contamination with DNA amplicons (see Methods and Supplementary Table 1). Moreover, the amplicon sequence of influenza A virus allowed us to distinguish between two different substrains we had used, namely A/Wy and A/WSN (Fig. 5) the latter of which is anticipated to circulate in future flu seasons[49]. Similarly, with a single primer pair we were able to distinguish between the three rhinoviral strains, namely HRV A1a, A1b, and A2 by polymorphisms in the respective amplicons. Taken together, our pipeline in its current form differentially detects seven different viral respiratory agents in a single reaction, contains various internal controls, a sample quality control, and by design has particularly high sensitivity for SARS-CoV-2. SARSeq thus represents a multiplexed, massively parallelized assay for saliva analysis by RNA sequencing to detect respiratory infections by means of RT-PCR and NGS.

## Discussion

Mass testing for SARS-CoV-2 by PCR-based methods with a focus on surveillance of asymptomatic individuals can help mitigate the effects of the COVID-19 pandemic, and this strategy has shown good results in China, South Korea, Taiwan, and Singapore[8–10]. Compared to antigen tests, which have also been effectively used for mass testing, e.g., in Slovakia[18], PCR-based assays are orders of magnitude more sensitive and this offers a number of advantages: infected individuals are detected with higher certainty, this makes it possible to pool samples, e.g., within families, and the increased robustness makes it compatible with self-sampling. Households or close groups will thus also be detectable as positive between peaks of viral titers of individual members. To further leverage these methods, the assays used for such large-scale testing must meet a number technical considerations: (i) the tests themselves must be highly specific to

avoid false positives leading to the isolation of individuals based on erroneous results; (ii) the costs for mass testing must be as low as possible to reasonably enable scaling; (iii) the assay must be massively scalable and return results in a short timeframe[50]; (iv) mass testing must not interfere with testing in medical/diagnostic facilities, it is thus preferable that it is neither carried out at the same facilities nor competes for supplies required to diagnose symptomatic patients.

Here, we described SARSeq, an NGS-based testing method that meets the technical considerations outlined above. The current design enables analysis of up to 36,000 samples in parallel, and we demonstrated the analysis of >18,000 samples in a single sequencing run. SARSeq shows high specificity at two levels: at the amplicon level it has maximal specificity as it detects the precise sequence of two independent SARS-CoV-2 amplicons; at the sample identification level, SARSeq employs a number of measures to completely suppress misassignment between samples. Moreover, the cost per sample (~2 Euros in our in-house pipeline) analyzed by SARSeq is low compared to other available tests; it relies on common reagents and enzymes that can be purchased at scale or produced in-house with standard biochemical methods. The costs of sequencing per sample also become negligible considering that they are divided over thousands of samples. Finally, SARSeq, with the exception of NGS, relies exclusively on equipment available in most molecular biology facilities in academia and industry, and does not compete for resources with other diagnostic tests.

Different SARS-CoV-2 detection assays have been optimized over the last few months, each with strengths and limitations. For SARSeq, a potential limitation is the time requirement of the assay. Two PCR reactions must be performed followed by NGS and analysis, so the theoretical minimum time required is around 15 h. In practice, our tests took at least 24 h from sample preparation to results. Therefore, SARSeq is not suited for situations where immediate results are required. In such cases, antigen tests[51] or RT-LAMP[52,53] are superior methods. Rather, SARSeq with its high throughput, sensitivity, and specificity is ideally suited for regular (e.g., once or twice a week) surveillance of infections in large organizations or populations. SARSeq might also be suitable to test symptomatic persons if a turnaround of 24 h for the test itself is acceptable. In addition, SARSeq can be implemented in epidemiology studies to understand the spreading dynamics of infections[54] and to investigate interaction between different pathogens across large populations[55]. However, the main advantage of SARSeq is that the same turnaround time of 15–24 h can be used to simultaneously test tens of thousands of samples. Therefore, SARSeq complements available diagnostic tests, increasing capacity to enable large-scale monitoring efforts.

Several alternative methods to detect SARS-CoV-2 by NGS have been developed. While some focus on viral genome sequencing and are thus of lower throughput[27,56], others aim to be used for detection of viral infection by amplicon sequencing at high throughput similar to SARSeq. In one strategy, samples are indexed during the RT step and PCR is performed in pool[26]. This approach has the advantage that early sample pooling circumvents the need for large numbers of individual PCR reactions. We anticipate, however, that such an approach would maintain the vast dynamic range in viral titers between samples and, as explained above, this leads to highly positive samples dominating the available NGS read space, thereby prohibiting true scalability while maintaining sensitivity.

In contrast, SwabSeq[22] and SARSeq use individual PCR reactions for each sample, which allows dynamic-range compression by endpoint PCR. In addition, both methods use a dual indexing strategy to gain the required robustness in sample recall that is key for diagnostic assays. However, SwabSeq and SARSeq differ

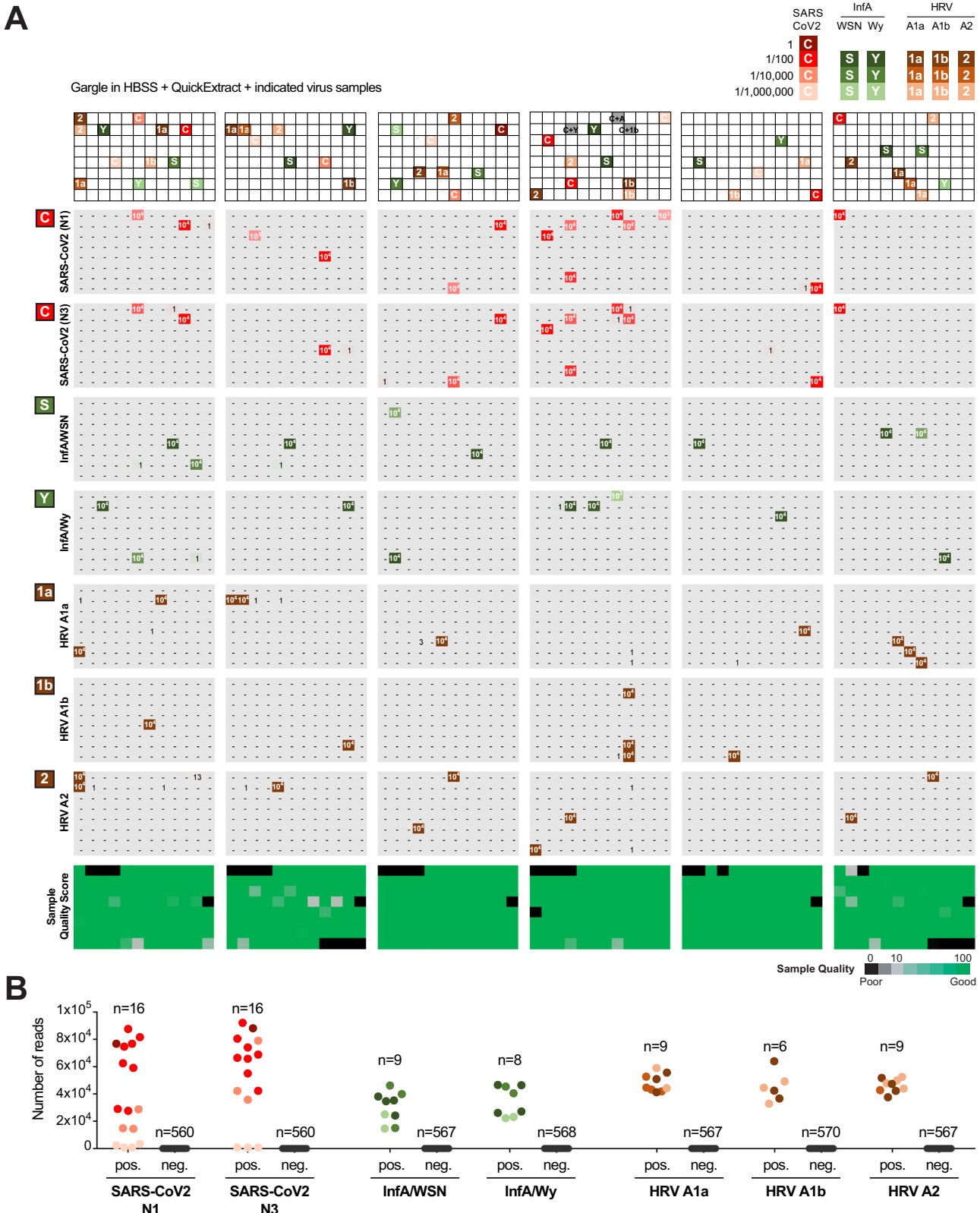

**Fig. 5 SARSeq can detect multiple respiratory viruses in a single reaction. A** Six 96-well plates filled with gargle in HBSS and inactivated in QuickExtract from our in-house pipeline were spiked with RNA from various respiratory viruses. For SARS-CoV-2, a positive gargle sample with Ct value of 30 was diluted as indicated. RNA for all other viruses was obtained from HEK cells 48 h after infection with the virus. Dilution indicated by voluminometric ratio. SARSeq was performed with six primer pairs in one reaction, namely N1 and N3 for SARS-CoV-2, Influenza A virus, human rhinovirus (HRV), and the ribosome. Influenza A substrains and HRV substrains were distinguished based on amplicon sequence variants. **B** Analysis of false-positive and false-negative rate of the experiment in panel **A**. As expected, 1:100,000 dilutions of the SARS-CoV-2 sample with a Ct of 30 were missed. All other positive samples were detected for all viral spike-ins.

in several important aspects that directly impact scalability and the multiplexed detection of different amplicons: SwabSeq uses a one-step RT-PCR reaction, whereas SARSeq performs RT and PCR in two steps, which we found to significantly suppress unspecific amplicons and thus make efficient use of the read space, a prerequisite for sensitivity, scalability, and multiplexing of amplicons (a one-step RT-PCR reaction is also compatible with SARSeq, albeit with slightly lower amplicon specificity, Fig. 1D). More importantly, due to a single indexing step, upscaling of SwabSeq is linear rather than combinatorial—every additional sample requires one additional primer pair per amplicon. In contrast, SARSeq uses a two-dimensional indexing strategy, which allows combinatorial (multiplicative) scaling of up to tens of thousands of samples with just a few hundred primer pairs and therefore leverages the sequencing capacities offered by NGS. By linear indexing, such dimensions would neither be cost-effective nor logistically feasible. Another important difference is that the amplicons generated by SwabSeq contain the flow cell adaptors but not the i5/i7 sequencing primer bindings sites and thus require a mix of custom sequencing primers, one for each amplicon. This limits the number of different amplicons that can be surveyed in parallel and also can cause heterogeneity in cluster signal intensity and thus impact sequencing quality. In contrast, all SARSeq amplicons contain standard i5/i7 sequencing primer bindings sites and are directly compatible with the regular sequencing protocols and reagents on all Illumina platforms. This facilitates the addition of further amplicons to the assay, since all are read out by the same standard i5/i7 binding sequencing primers. Therefore, the extra PCR reaction that SARSeq requires is a small technical burden, which however allows dramatic improvements in scalability of samples and amplicons and yields amplicons with homogenous sequencing properties.

In practical terms the limitation of SARSeq is defined by the number of available PCR machines and pipetting robots. One PCR machine can process around 1500 samples per day, so the investment costs are negligible if sample throughput is high. To process 10,000 samples in 1 day about seven persons and ten PCR machines are needed. SARSeq thus solves the technical challenges downstream of sample acquisition, shifting the bottleneck toward truly high-scale testing from the actual assay to developing matching logistics for sample collection, maintaining supply chains, developing appropriate data management tools, and also often overcoming legal hurdles. However, we envision ways in which SARSeq can be implemented right away, to already significantly contribute to detecting infection events before they spread. In the first, samples can be collected and inactivated locally, potentially also first PCR might be performed using prepared and distributed primer arrays, followed by shipment to a centralized location for PCR2, sequencing, and analysis. Importantly, depending on the legal situation and aim, this pipeline does not necessarily need a human diagnostics lab and would thus not block important infrastructure. In the second, companies, universities, other types of institutions could implement a regular sample collection strategy among employees/students/other members and team up with a local academic or industry lab that can with relatively little effort implement this protocol.

Beyond SARS-CoV-2, SARSeq allowed the detection of other respiratory RNA viruses in parallel. Specifically, we used SARSeq to detect influenza A virus, influenza B virus, and HRV and this list can be easily expanded to additional infectious agents, both circulating and also newly emerging pathogens. Detection limits for these pathogens must still be calibrated in clinical samples. Moreover, SARSeq is not limited to respiratory specimens, but we envision that this pipeline could be used for other human and animal samples or even monitoring pipelines such as those that sample wastewater. PCR-based tests are immediately adaptable to the detection of new amplicons or viral variants and SARSeq provides a platform and an experimental outline to do so. We therefore think that SARSeq represents a first-line mass testing strategy in the event of future epidemic outbreaks.

The current pandemic has challenged healthcare systems worldwide, but it has also led to an unprecedented concerted response on multiple levels. We wish to contribute to the fight and therefore welcome anyone interested in implementing this method to contact us.

## Methods

**Sample material and ethics**. The present study includes preliminary investigations and results as a basis for a clinical performance study approved by the local Ethic Committee of Vienna (#EK 20-208-0920). For that, left-over samples from healthy participants were obtained from an anonymous routine SARS-CoV-2 screening pipeline, and left-over patient samples in Fig. 4G, H were obtained by the Austrian Agency for Health and Food Safety in a diagnostic pipeline and provided to us fully anonymized. For VTM samples used for Fig. 4A–F, an additional approval (#06-04-9-33163 from July 21, 2020) was obtained from the Ethics Committee of the Clinical Center of the University of Sarajevo. The study was conducted in accordance with the Declaration of Helsinki.

**Input sample preparation**. The pipeline we describe can start from a variety of different input samples. The types of samples we have tried are:

Purified RNA from gargle samples
Gargle samples mixed with DNA QuickExtract solution from Lucigen (1:1 ratio)
Swabs in VTM mixed with DNA QuickExtract (1:1 ratio)

Samples mixed directly with QuickExtract were incubated for 5 min at 95 °C for inactivation and used directly or stored frozen at –80 °C. We did not observe a decline in positive signals upon freezing and thawing (even after two cycles of freeze-thaw).

The samples were arrayed in 96-well plates. The described reaction setup uses up to 5 µL of any of the above described samples.

**Reverse transcription**. RT was performed with reverse transcriptase, homemade Ribonuclease inhibitor and a primer mix containing random hexamers as well as two 12-mer oligonucleotides that prime on the SARS-CoV-2 N gene.

A master mix containing all components listed below was prepared and distributed to 96-well plates (20 µL per well). Using a liquid-handling robot (or multi-channel pipettes), 5 µL of each sample was transferred to each individual well containing the RT reaction mix. RT reactions were set up at room temperature. Plates were sealed with aluminum sealing foil (facilitates easy removal after RT reaction that reduces vibrations in wells avoiding generation of aerosols that may cause cross-contamination between samples) and incubated in a thermocycler following conditions listed below.

Master mix composition per reaction/well (volumes in µL)

| | |
|---|---|
| 10× RT Buffer[A] | 2.5 |
| 25 mM each dNTP | 0.5 |
| 1 M DTT | 0.1 |
| RT primer mix[B] | 2.0 |
| Ribonuclease inhibitor | 0.5 |
| Reverse transcriptase | 0.5 |
| Water | 13.8 |

Wherever mentioned, for each reaction 1000 copies of Ribosome synthetic RNA spike-in and 50 copies of each N1 and N3 RNA synthetic spike-in were included in the RT reaction master mix. Also, wherever mentioned Thermo Fisher/Invitrogen™ SuperScript™ III or Luna Universal One-Step RT-qPCR Kit (NEB) or homemade reverse transcriptase 2.5 (see below for details) was used for RT. In all other experiments, homemade reverse transcriptase 3 was used for RT.

Thermocycler program:
5 min at 25 °C (primer annealing)
15 min at 55 °C (reverse transcription/RT was carried out at 42 °C for reverse transcriptase 2.5)
3 min at 95 °C (RT inactivation)
Cool down to 12 °C (removing the plate while it is still hot will cause bending of the plastic, making further pipetting and sealing more difficult)
[A]10× RT Buffer composition:
200 mM Tris-HCl pH 8.3
500 mM KCl
50 mM $MgCl_2$

200 mM $(NH_4)_2SO_4$
1% Triton X-100
[B]RT primer mix composition: 12.5 mM of each random hexamer, N gene-specific 12-mer #1(AACCAAGACGCA) and N gene-specific 12-mer #2 (GGTGGGAATGTT). Final concentration of RT primer mix in the complete 25 µL RT reaction is 1 mM each.

*In vitro transcription of spike-in synthetic RNA templates for reverse transcription controls.* For RTC, gBlock was obtained from IDT. Using IDT, synthetic template RTC was PCR amplified and cloned into pCR2.1 plasmid by TOPO cloning. For cloning N1 spike-in N1FF, N1FR, N1FR, N1RR, insertF, and insertR, oligos were annealed and cloned into pCR2.1 plasmid by ligation at SpeI and EcoRI sites. Similarly, for cloning N3 spike-in N3FF, N3RF, N3FR, N3RR, insertF, and insertR, oligos were annealed and cloned into pCR2.1 plasmid by ligation at SpeI and EcoRI sites. RTC gBlock sequence and oligo sequences used to clone N1 and N3 spike-in templates are given in Supplementary Data 2. Spike-in template containing plasmid clones are confirmed with Sanger sequencing. For efficient in vitro transcription, plasmids were linearized downstream of the T7 promoter and spike-in template by cutting with a unique restriction enzyme. In vitro transcription was carried out using NEB HiScribe™ kit according to the manufacturer's instructions. Transcribed reactions were treated with Turbo DNAse/Thermo Fisher for 1 h and RNA is purified using Zymo RNA clean and concentrator spin columns. RNA was aliquoted and stored at –80 °C. For oligonucleotide sequence please refer to Supplementary Data 2 tab "oligo sequence for T7 transcripts."

**First PCR (sample indexing).** A master mix containing all components listed below, including homemade Hot Start Taq Polymerase and Uracil DNA glycosylase (Antarctic Thermolabile UDG from NEB), was prepared and distributed to a deep-well 96-well plate. The 96-primer pair combinations[C] containing dual well barcodes were also arrayed in 96-well plates (multiple primer plates can be prepared simultaneously and stored frozen at –20 °C). Using a liquid-handling robot, the 96 sets of barcoded primers were added to the PCR master mix and mixed thoroughly. Then, 25 µL of this complete 2× PCR mix was added to the 25 µL RT reactions prepared as above. Plates were sealed with aluminum sealing foil and incubated in a thermocycler following the conditions listed below.

All components were kept at room temperature during reaction set up; together with the first step in the thermocycler, a 10-min incubation at 30 °C, this provides the right conditions for UDG to act on Uracil-containing amplification products of previous PCR reactions, thereby removing spurious carry over contaminants. After UDG heat inactivation, the subsequent PCR reaction was again carried out in the presence of UTP to prevent carry over contamination in following runs.

Master mix composition per reaction/well (volumes in µL)

| | |
|---|---|
| 10× PCR Top Up Buffer[D] | 2.5 |
| 100 mM dUTP | 0.07 |
| Hot Start Taq Polymerase | 0.5 |
| Antarctic thermolabile UDG | 0.5 |
| Water | 16.43 |

Thermocycler program:
10 min at 30 °C (for high UDG activity)
3 min at 95 °C (UDG inactivation and Hot Start Taq activation)
45 cycles of: 20 s at 95 °C, 30 s at 58 °C, 20 s at 72 °C
2 min at 72 °C
Cool down to 12 °C (removing the plate while it is still hot will cause bending of the plastic, making further pipetting and sealing more difficult)
[C]PCR primer mix composition: 2 mM of each forward and reverse primer, for all viral amplicons and 1 mM of each forward and reverse primer for the rRNA amplicon (final concentration of each primer pair in the complete 50 µL reaction was 200 and 100 nM, respectively)
[D]10× PCR Top Up Buffer composition:
750 mM Tris-HCl pH 8.3
200 mM $(NH_4)_2SO_4$
1% Triton X-100

**Plate pooling.** All well-barcoded PCR products from a single 96-well plate were pooled, typically 20 µL of each reaction was combined in a plastic reservoir using a multi-channel pipette, and after mixing thoroughly 1 mL was transferred to an Eppendorf tube. This was repeated for every PCR plate. Then, 5 µL from each plate pool was re-arrayed in a new 96-well plate and treated with 2 µL of illustra Exo-ProStar 1-step for 30 min at 37 °C followed by 15 min at 80 °C to remove any leftover primer.

**Second PCR (plate indexing and addition of sequencing adaptors).** A master mix with all components listed below was distributed across a 96-well plate (37.5 µL/well). To each we added 10 µL of unique dual-indexed i5/i7 primer pairs (Custom synthesized index primers with Nextflex barcodes, arrayed in 96-well plates) and 2.5 µL of ExoProStar-treated PCR1 pool. The reactions were run for eight cycles to add sequencing adaptors with plate barcodes.

Master mix composition per reaction/well (volumes in µL)

| | |
|---|---|
| 10× Sequencing-ready PCR Buffer[E] | 5 |
| 25 mM each dNTPs | 0.5 |
| 100 mM dUTP | 0.07 |
| Hot Start Taq Polymerase | 0.5 |
| Water | 31.43 |

Thermocycler program:
3 min at 95 °C
8 cycles of: 15 s at 95 °C, 30 s at 65 °C, 30 s at 72 °C
2 min at 72 °C
Cool down to 12 °C
[E]10× Sequencing-ready PCR Buffer composition:
750 mM Tris-HCl pH 8.3
200 mM $(NH_4)_2SO_4$
20 mM $MgCl_2$
0.1% Tween-20

**Pooling and preparation for sequencing.** All samples from a 96-well plate (20 µL from each well) were pooled and 250 µL of pooled sample was resolved on a 2% agarose gel and 220–260 bp amplicons were excised and gel purified using Qiagen gel extraction kit.

To ensure fast turnaround, the preparation of libraries for Illumina sequencing was optimized empirically. In the first four sequencing runs, standard quality control of the library, including Qubit measurement, a size analysis and qPCR, was performed. A correlation between the concentration measurement and the qPCR was detected. In every case the molarity determined by qPCR was 10× higher than the concentration measured by Qubit. Thus, we were able to omit the size analysis and the qPCR, which are both time consuming. The library concentration was determined by three independent Qubit measurements, the obtained value in ng/µL was multiplied by 10 and used as the molarity of the sample in nanomolar. This procedure enables us to start the sequencer within 15 min after receiving the sequencing library. Final preparation of the sequencing run happens according to Illumina's guidelines, including denaturation of the sample, neutralization and final dilution for sequencing.

**Sequencing.** Depending on the sequencer type, the following concentrations were used for sequencing: 10 pM for MiSeq V2 chemistry, 15 pM for MiSeq V3 chemistry, 2.2 pM for NextSeq550 high output, and 1.3 pM for NextSeq550 medium output. In every sequencing run 10% of PhiX library were spiked-in to increase complexity. To avoid contaminations with barcodes from previous sequencing runs, the sequencers were washed with bleach according to Illumina's guidelines before every run[22]. In addition, that to avoid cross-contamination of barcodes from previous runs, in practice, even if running a smaller number of samples, having 384 plate barcodes (second dimension) allowed us to alternate the subsets of indices used and thereby filter against any DNA remnants from previous runs that might be in the sequencer.

**Data analysis.** The NGS data (fastq.gz files) were mapped in a single pass to sets of expected amplicon sequences and to sets of expected well- and plate-indices using dedicated shell and awk scripts based on string-hashing that allows for 0 or 1 mismatch per amplicon and index. During method development, different parameters were tested and optimized, including single- versus paired-end sequencing, the sequencing platforms (MiSeq vs. NextSeq), and the exact positions of the indices in the primers (and thus in the reads) and the analysis was adjusted accordingly (the analysis script we make available is compatible with the final primer- and parameter set recommended for use). For redundant dual indexing, we required the correct redundant encoding of plate and well. The i5 and i7 index reads signify the plate-indices, and parts of the forward and reverse reads (in the case of paired-end sequencing) signify the well-indices. As the well-index in the forward read starts at random offsets, we first determine the amplicon identity and position, then infer the position of the well index, and finally compare the well index to the valid well index pairs; all reads with invalid plate- or well-index pairs were excluded. For the final set of primers, the offsets are made consistent for all amplicons of a given well, changing between 1 and 4 between wells such that the well-index starts between positions 2 and 5.

The analysis script is available on GitHub at https://github.com/alex-stark-imp/SARSeq and at https://starklab.org.

*Viral amplicons.* Supplementary Data 2, tab "viral amplicons," contains amplicons that were extracted from the NGS reads: lower case denotes primer sequences, upper case fonts are specific amplicon sequences.

**Quantitative RT-PCR (qPCR) assay.** RT was performed as above. For qPCR analysis results in Supplementary Figs. 2A and 3C, 2 μL of first strand cDNA was taken for qPCR analysis in Promega GoTaq® qPCR Master Mix (Ref: A6001). Reactions were run at 95 °C for 3 min, followed by 50 cycles of 95 °C for 15 s, 58 °C for 30 s, and 72 °C for 20 s with melt curve analysis in the end in a BioRad CFX Connect™ Real-Time System. For all N1 TaqMan qPCR analysis results, RT was performed as above in RT section by taking 5 μL of sample in 25 μL RT reaction. For qPCR analysis, 25 μL of PCR1 top up reaction mix with 1.5 μL of CDC-N1 primer/probe set (IDT 10006713/sub part 10006600) was added to above 25 μL RT reaction mix. Reactions were run at 95 °C for 3 min, followed by 45 cycles of 95 °C for 15 s and 55 °C for 45 s in a BioRad CFX Connect™ Real-Time System.

**Expression and purification of homemade enzymes**
*Reverse transcriptase 2.5.* pET15b-His6-Reverse transcriptase 2.5 was transformed into competent Rosetta 2 (DE3) *E. coli* cells and plated on LB plates containing Ampicillin (50 μg/mL). A culture (LB+50 μg/mL) was inoculated with a single colony and incubated at 37 °C overnight with 220 rpm agitation on an orbital shaker.

Two liters of pre-warmed LB (+50 μg/mL ampicillin) was inoculated with 15 mL of the overnight culture in a baffled 5-L bacterial culture flask. The cells were grown (37 °C/180 rpm agitation) until an OD600 of 0.7, transferred to a 15 °C pre-cooled shaker and induced with 0.5 mM IPTG after 30 min. The next day the cells were harvested by centrifugation (5000 × g) and the pellet was resuspended 1:5 (w/v) in Resuspension buffer (10 mM NaPi pH 7.5, 500 mM KCl, 20 mM (NH₄)₂SO₄, 1 mM beta-mercaptoethanol, 0.01% NP-40, 5% glycerol, 10 mM imidazole). The resuspension was sonicated in a glass rosette with a Branson Digital Sonifier (4 min, 60% amplitude, 1-s on/1-s off). The lysate was cleared with 40,942 × g at 4 °C for 20 min. The supernatant was applied to a HisTrap 5 mL HP (GE Healthcare) column and washed with IMAC Buffer A (20 mM HEPES pH 7.5, 300 mM KCl, 20 mM (NH₄)₂SO₄, 1 mM beta-mercaptoethanol, 0.01% NP-40, 5% glycerol). Bound proteins were eluted with IMAC Buffer A supplemented with 250 mM imidazole. The eluate was diluted with a low salt buffer (20 mM HEPES pH 7.5, 20 mM (NH₄)₂SO₄, 1 mM beta-mercaptoethanol, 0.01% NP-40, 5% glycerol, 10 mM imidazole) to a final conductivity of 12 mS/cm. The protein was applied onto a manually packed SP Sepharose FF (GE Healthcare) 20 mL column equilibrated in cation exchange buffer A (20 mM HEPES pH 7.5, 20 mM (NH₄)₂SO₄, 1 mM TCEP, 0.01% NP-40, 5% glycerol). The protein was eluted with a linear gradient from 50 to 500 mM KCl in cation exchange buffer A in 20 CVs. The main peak fractions were diluted to a maximum conductivity of 12 mS/cm with 20 mM HEPES pH 7.5, 0.01% NP-40 and 5% glycerol. The final concentrating step included re-application of the eluate onto the cation exchange column and a one-step elution with 300 mM KCl in cation exchange buffer A. Finally, 99.9% glycerol was added to the eluate under constant low speed stirring to reach a final glycerol concentration of 50%. Additional NP-40 was added to reach a final concentration of 0.01% (v/v). The final storage buffer contained 10 mM HEPES pH 7.5, 150 mM mM KCl, 10 mM (NH₄)₂SO₄, 0.5 mM TCEP, 0.01% NP-40 and 50% glycerol. Protein concentrations were determined by absorbance at 280 nM wavelength.

*Ribonuclease inhibitor.* pET28b-His6-RNase Inhibitor from *Mus musculus* was transformed into competent Rosetta 2 (DE3) cells. Cells were grown in LB medium supplemented with 50 μg/mL Kanamycin to an OD600 of 0.7 and expression was induced with 0.5 mM IPTG and carried out overnight at 15 °C. The cells were harvested by centrifugation with 5000 × g at 4 °C, resupended five times (w/v) in Lysis buffer (50 mM NaPi pH 8.0, 500 mM KCl, 0.5 mM TCEP, 0.01% NP-40 and 10% glycerol) and sonicated with a Branson Digital Sonifer (4 min, 60% amplitude, 1-s on/1-s off). The cell lysate was cleared by centrifugation with 40,942 × g at 4 °C for 20 min. The cleared cell lysate was applied to a HisTrapHP (GE Healthcare) column equilibrated in IMAC A buffer (20 mM HEPES pH 8.0, 500 mM KCl, 1 mM beta-mercaptoethanol, 0.01% NP-40, 5% glycerol), washed with 10% IMAC B buffer (20 mM HEPES pH 8, 500 mM KCl, 1 mM beta-mercaptoethanol, 0.01% NP-40, 5% glycerol, 300 mM imidazole) over 5 CV and eluted with 100% IMAC B buffer. The protein was diluted to final conductivity of 12 mS/cm and applied to a ResourceQ (GE Healthcare) column equilibrated in IEX A buffer (20 mM HEPES pH 7.5, 50 mM KCl, 0.01% NP-40, 5% glycerol). The protein was eluted with a linear gradient from 50 to 500 mM KCl in IEX A buffer over 30 CV. The His-tag cleavage was carried out overnight with 3C Protease at 4 °C and removed with Ni-NTA agarose beads. The supernatant containing tag-free RNase Inhibitor was applied to a Superdex 75 (GE Healthcare) column equilibrated in SEC buffer (40 mM HEPES pH 7.5, 100 mM KCl, 1 mM TCEP, 0.01% NP-40, 5% glycerol). Finally, the protein was diluted with 99.9% glycerol to a final glycerol concentration

of 50%. Ribonuclease inhibitor expression and purification protocol is optimized based on previous study[57].

*Hot Start Taq polymerase.* Taq polymerase expression plasmid is transformed into *E. coli* strain DH5 alpha and plated on LB plate containing Ampicillin. After selection, inoculated a single colony of *E. coli* into 5 mL of LB medium with 100 μg/mL ampicillin and incubated in orbital shaker at 37 °C until a visible turbidity. Transferred this inoculum into 100 mL LB+ Ampicillin and incubated for overnight in orbital shaker at 37 °C. Next day, inoculated overnight culture into 2 L of pre-warmed LB + Ampicillin and incubated on orbital shaker at 37 °C until OD₆₀₀ reaches 0.5–0.6. Protein expression is induced by adding IPTG to 1 mM final concentration and incubated in orbital shaker at 37 °C for an extra 3 h. Cells were harvested by centrifugation at 4218 g/15 min/4 °C and washed the cell pellet with 1X PBS. Resuspended cell pellet in 100 mL of Buffer A (25 mM HEPES-KOH pH 7.5, 25 mM Glucose, 200 mM KCl, 1 mM EDTA, 0.5% Tween-20, and 0.5% NP-40) with Protease inhibitors (Roche). Incubated above suspension in a 250 mL Erlenmeyer flask for 1 h at 75 °C. Pelleted cell debris by centrifugation at 30 min at 4 °C/74,766 × g and collected supernatant. Equilibrated DE-52 (DE-52, pre-swollen form, Whatman) or DEAE resin (BioRad #156-0021) by washing it three to four times with Buffer A and centrifuged 2 min at 3739 × g at 4 °C. Batch incubated the supernatant with DE-52 or DEAE for 15 min at 4 °C (resin should not settle down). Centrifuged suspension for 2 min at 3739 × g at 4 °C and collected the supernatant. DE-52 or DEAE resin is washed one time with 100 mL Buffer A, centrifuged for 2 min at 3739 × g at 4 °C and collected the supernatant. Both supernatant fractions are combined and diluted to 40 mM KCl with Buffer B (20 mM HEPES-KOH pH 7.5, 1 mM EDTA, 0.5% Tween-20, and 0.5% NP-40). Applied the supernatant on a Poros 20 CM 16 mmD/100 mmL column (this column is not produced any longer but can be replaced by any strong cation exchange column). Before loading the samples, equilibrated the column with 40 mM KCl. After applying the sample, washed the column with 40 mM KCl, till there is a stable baseline. Step eluted the Taq polymerase with 300 mM KCl in buffer B. Collected the peak fractions and dialyzed in dialysis buffer (20 mM HEPES-KOH pH 7.5, 100 mM KCl, 50% glycerol, 1 mM EDTA, 0.5% Tween-20, and 0.5% NP-40 and 1 mM DTT) for overnight at 4 °C (volume of protein solution will be reduced about 1/3 ratio). After dialysis, Taq polymerase is aliquoted and stored in the freezer by snap freezing (for long term, store at –80 °C and for short term, store at –20 °C). Measured the activity of Taq polymerase and diluted accordingly with dialysis buffer. To prevent unspecific primer elongation during setting up the PCR reaction we followed a published protocol[58]. Specifically, an aptamer that blocks polymerase activity at room temperature was added to the TAQ polymerase stocks and used at 100 nM in the PCR reaction. The aptamer is modified with phosphotionate nucleotides at the 5′end and a C3-spacer at the 3′end to protect for exonucleolytic degradation and to prevent undesired elongation of the aptamer itself.

Aptamer sequence: TGGCGGAGCAAGACCAGACAATGTACAGTATTGGC
CTGATCTTGTGTATGA

**Generation of figures.** Statistical analysis and plots were done using GraphPad Prism 8.4.3., plate layouts were illustrated in Microsoft Excel, and figures were assembled in Adobe Illustrator CS6.

**Reporting summary.** Further information on research design is available in the Nature Research Reporting Summary linked to this article.

## Data availability
Sequencing data for all experiments are available at GEO, under accession number GSE163688. Figures with associated raw data are: 1D–E, 2, 3B–E, 4A–H, 5, S1C, S1D, S2B, S3, S4, S5, S6, S7, and S8. All sequencing data are available at GEO, all primer sequences are submitted as Supplementary Data 2. Source data are provided with this paper: https://www.ncbi.nlm.nih.gov/geo/query/acc.cgi?acc=GSE163688.

## Code availability
Custom code was used to analyze all NGS data. The script is available on GitHub at https://github.com/alex-stark-imp/SARSeq and at https://starklab.org.

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

## Acknowledgements

In immediate response to the start of the pandemic, many people in our campus united to contribute with their skills in molecular biology toward the developing of various testing strategies. These acknowledgements inevitably fall short in thanking everyone involved, but we wish to highlight those who made the biggest contributions. In particular we thank Stefan Ameres, Julius Brennecke, Harald Isemann, Andrea Pauli, and Johannes Zuber for close interaction and support throughout. Harald Isemann coordinated the strong administrative and financial support from our institutes to these endeavors, for which we are extremely grateful. We thank the entire RT-LAMP team especially Max Kellner and Martin Matl for their close interaction and support in various aspects, Andrea Pauli and Julius Brennecke also for their tremendous help in obtaining patient samples. Various people invested their time and effort to implement the in-house testing pipeline that enabled us to safely come back to work (Johannes Zuber, Harald Scheuch, Peter Steinlein, Stefan Ameres, and Sabina Kula), and also provided samples for this work (Katharina Bergauer, Martina Weissenböck, and Barbara Werner). We are extremely grateful to Irma Salimović-Bešić and Sandra Vegar-Zubovic for providing patient samples in difficult days. We are indebted to Tim Clausen and Anton Meinhart for supporting enzyme purification efforts, to Andreas Sommer, Ido Tamir, and the entire NGS team at the VBCF. We also thank Raphael Manzenreither for technical help, Gijs Versteeg for virus preparations and other help, and Edith Soucek for support. We thank Katrina Woolcock from Life Science Editors for editing, as well as the community of COVID fighters worldwide. We thank all employees for consent to sample use and ethics boards for rapid processing of the requests. Last but not least we would also like to thank our labs and all coworkers for understanding and supporting us in so many ways. R. Y. was partially supported by Oliver Bell's laboratory at USC. Curiosity-driven biomedical research at the IMP is largely sponsored by Boehringer Ingelheim. IMBA is generously funded by the OEAW.

## Author contributions

R. Y., L. C., and U. E. performed experiments with help of E. Ö. and M. S. Experimental design was by R. Y., A. S., L. C., and U. E. NGS analysis was performed by A. S. A. B. set up and performed automated pipetting. A. V. supported optimization of RT and PCR conditions, oversaw and executed NGS. R. H. shared reagents, expertise, and supported experimental design. K. U., B. H., and D. K. purified proteins used in this study. J. B. cloned various constructs and provided technical help. E. S., A. K-K., and S. I. provided clinical samples and ethics approval. A. Z., T. S., M. F., J. S., P. H., and F. A. provided ethics and samples and results from a human diagnostics lab. The VCDI is a consortium of multiple people that established various SARS-CoV-2 testing pipelines in close collaboration and supported sample acquisition as well as multiple steps of optimizations and upscaling. F. A. oversaw the human diagnostics pipeline, assessed results, and corrected the manuscript. Figures were generated by L. C. and U. E. A. S., L. C., and U. E. supervised the study, and wrote the manuscript.

## Competing interests

U. E., L. C., A. S., and R. Y. declare the following competing interest: a European patent application EP 20202627.4 was filed on October 19, 2020. We will, however, grant non-exclusive licenses for all non-commercial use. U. E. consults for Tango Therapeutics and is a co-founder of JLP Health. All other authors declare no competing interests.

## Additional information

## VCDI

Mariam Al-Rawi[12], Stefan Ameres[1,12], Juliane Baar[1], Benedikt Bauer[2], Nikolaus Beer[1,2,4], Katharina Bergauer[2], Wolfgang Binder[12], Claudia Blaukopf[1], Boril Bochev[1,2,4], Julius Brennecke[1], Selina Brinnich[3], Aleksandra Bundalo[2], Meinrad Busslinger[2], Aleksandr Bykov[2], Tim Clausen[2,11], Luisa Cochella[2✉], Geert de Vries[1], Marcus Dekens[2], David Drechsel[3], Zuzana Dzupinkova[1,2,4], Michaela Eckmann-Mader[3], Ulrich Elling[1✉], Michaela Fellner[2], Thomas Fellner[3], Laura Fin[2], Bianca Valeria Gapp[1], Gerlinde Grabmann[3], Irina Grishkovskaya[2], Astrid Hagelkruys[1], Bence Hajdusits[2], Dominik Handler[1], David Haselbach[2], Robert Heinen[1,2,4], Louisa Hempel[3], Louisa Hill[2], David Hoffmann[1], Stefanie Horer[2], Harald Isemann[2], Robert Kalis[2], Max Kellner[1,2], Juliane Kley[2], Thomas Köcher[3], Alwin Köhler[12], Darja Kordic[2], Christian Krauditsch[1], Sabina Kula[1,2,4], Sonja Lang[3], Richard Latham[2], Marie-Christin Leitner[1], Thomas Leonard[12], Dominik Lindenhofer[1], Raphael Arthur Manzenreither[1], Martin Matl[1], Karl Mechtler[2], Anton Meinhart[2], Stefan Mereiter[1], Thomas Micheler[3], Paul Moeseneder[1], Tobias Neumann[2], Simon Nimpf[2], Magnus Nordborg[4], Egon Ogris[12], Ezgi Özkan[2], Michaela Pagani[2], Andrea Pauli[2], Jan-Michael Peters[2,11], Petra Pjevac[13,14], Clemens Plaschka[2], Martina Rath[2], Daniel Reumann[1], Sarah Rieser[2], Marianne Rocha-Hasler[13], Alan Rodriguez[1,2], Nathalie Ropek[12], James Julian Ross[1], Harald Scheuch[1,2,4], Karina Schindler[2], Clara Schmidt[1], Hannes Schmidt[13], Jakob Schnabl[1], Stefan Schüchner[12], Tanja Schwickert[2], Andreas Sommer[3], Daniele Soldoroni[3], Johannes Stadlmann[15],

Alexander Stark [2,11], Peter Steinlein[1,2,4], Marcus Strobl[1], Simon Strobl[3], Qiong Sun[2], Wen Tang[2], Linda Trübestein[12], Johanna Trupke[3], Christian Umkehrer[2], Sandor Urmosi-Incze[3], Kristina Uzunova[1,2,4], Gijs Versteeg[12], Alexander Vogt[3], Vivien Vogt[2], Michael Wagner[13,14], Martina Weissenboeck[2], Barbara Werner[3] & Ramesh Yelagandula [1] & Johannes Zuber[2,11]

[12]Max Perutz Labs, Medical University of Vienna, Vienna Biocenter (VBC), Vienna, Austria. [13]Centre for Microbiology and Environmental Systems Science, University of Vienna, Vienna, Austria. [14]Joint Microbiome Facility of the University of Vienna and Medical University of Vienna, Vienna, Austria. [15]Institute of Biochemistry, University of Natural Resources and Life Sciences (BOKU), Vienna, Austria.

