## [Peer Review File · Nature Communications]

Reviewers' Comments:

Reviewer #1:

Remarks to the Author:

Yelagandula et al. present a high-throughput method, SARSeq, for the detection of SARS-CoV-2 RNA and other respiratory RNA viruses in parallel from gargle and swab samples without RNA extraction. The benefit of the method is that it uses PCR machines instead of light cyclers and then sequences PCR products on an Illumina machine. The authors convincingly solve most of the challenges with such a testing setup and demonstrate in a proof of principle how such an approach can be used for surveillance of thousands of samples.

Major comments:

- One major technical challenge in Illumina sequencing that we encounter often in our lab is that sequences from one Illumina run can be seen in subsequent five or more runs due to contamination not only present in the Illumina machine that can be treated with bleach, but also due to contamination present in the lab where samples are prepared for sequencing or where the sequencers are run. This could make a problem for SARSeq in that a truly negative sample could be declared as a positive one. I would like the authors to check if they see sequences from previous runs appear in five subsequent runs and give us those numbers. Since this is a possibility, authors could discuss how to solve that problem, for example by keeping a journal of positive index combinations so that one could exclude those in future runs.
- Authors give a false impression that SARSeq is as sensitive as the gold-standard RT-qPCR whereas, in reality, SARSeq is as sensitive as RT-qPCR if there is no RNA-extraction done. However, gold-standard diagnostic RT-qPCR is always done with RNA extraction which allows for two orders of magnitude higher limit of detection (10 cp/ul vs. 0.1 cp/ul in this assay, compare <https://jcm.asm.org/content/58/9/e01535-20> tables 1 and 2 with Figure 3B in this paper). I think the authors should discuss this.
- How many hours have PCR machines run in order to do RT and PCR of 10,000 samples? How many machines were used? Running all those machines sounds like a logistical challenge and should be mentioned. How many hours and PCR machines would it take from gargle collection to declaration of positive samples for 10,000 samples? It would be useful to have estimates of the time that machines and people would need to process 10,000 samples from collection to end result (positive or negative sample) since time and machines could be prohibiting from testing so many samples.

Minor comments:

- First sentence in the last paragraph on page 2 is not complete, please revise.
- "This pipeline also suppressed read misassignment across plates, such that the highly-positive positions of Fig. 3A, C, E were not detected on other plates run in the same experiment" this sentence references a wrong figure.
- Please give a reference for this claim: "It is thought that infectious COVID-19 patients show viral titers of $>10^3/\mu\text{l}$ ". Is this number for gargles (of how many ml?) or swabs (of how many ml?).
- "9 out of 30 plates are shown in Fig. 3E", here Fig. 3F is meant.

Reviewer #2:

Remarks to the Author:

This manuscript by Yelagula et al. describes the development of SARSeq, an NGS-based SARS-CoV-2 diagnostic assay that allows to multiplex thousands of samples in one sequence run.

SARSeq shows a lot of similarities with SwabSeq, but has added a few innovations, including testing of saliva samples, and an innovative method to barcode individual samples. In addition, SARSeq allows for the testing of multiple respiratory pathogens in one assay.

This reviewer has a few points that need to be addressed:

- The manuscript is very technical, and it is sometimes difficult to follow, because minuscule details are highlighted that distract from the main message. The manuscript should also be shortened, and a lot of the details should go to the supplementary data.
- One of the strong points of SARSeq, the detection of multiple respiratory pathogens, requires more attention. First, the vast majority of the experiments were conducted in samples with only one pathogen. I'm worried that for samples that have both a low titer and high titer pathogen, the sensitivity of the low titer pathogen is affected. Secondly, the influenza and rhinovirus assays should be validated against a clinical qPCR assay (similar to the experiments done for SARS-CoV-2).
- The authors should deposit the final protocol in a repository like protocols.io, github or something similar. Also, the SARSseq website you are referring to in the manuscript does not have any content.

REVIEWER COMMENTS

Reviewer #1 (Remarks to the Author):

Yelagandula et al. present a high-throughput method, SARSeq, for the detection of SARS-CoV-2 RNA and other respiratory RNA viruses in parallel from gargle and swab samples without RNA extraction. The benefit of the method is that it uses PCR machines instead of light cyclers and then sequences PCR products on an Illumina machine. The authors convincingly solve most of the challenges with such a testing setup and demonstrate in a proof of principle how such an approach can be used for surveillance of thousands of samples.

We thank the reviewer for this overall positive assessment and the good summary of the major benefits. Indeed, SARSeq allows the use of an available infrastructure outside testing labs for the large-scale detection of infections.

Major comments:

- One major technical challenge in Illumina sequencing that we encounter often in our lab is that sequences from one Illumina run can be seen in subsequent five or more runs due to contamination not only present in the Illumina machine that can be treated with bleach, but also due to contamination present in the lab where samples are prepared for sequencing or where the sequencers are run. This could make a problem for SARSeq in that a truly negative sample could be declared as a positive one. I would like the authors to check if they see sequences from previous runs appear in five subsequent runs and give us those numbers. Since this is a possibility, authors could discuss how to solve that problem, for example by keeping a journal of positive index combinations so that one could exclude those in future runs.

We are grateful to the reviewers for bringing up this very important point. Contamination at different levels is indeed something we are aware of and were extremely careful about while setting up experiments. Specifically, we set up pre- and post-amplification labs that are physically separated into different floors/buildings, and also separated from the sequencing facility. As a further measure we use different sets of i5/i7 indices between consecutive runs to further minimize such problems. This is enabled because our indexing strategy of up to 36000 distinct combinations, which relies on 384 different i5/i7 index pairs is, in practical terms, not completely covered within a single run.

In addition, based on previous alerts by the Swab-seq team (Bloom et al, medRxiv, 2020), the sequencers were bleached before running. This is particularly important for all flow cells except HiSeq, namely MiSeq, NextSeq500/2000, NovaSeq, iSeq, MiniSeq, where tubing is not exchanged between independent runs. However, even with all these precautions it is possible (although we have not observed this in our >15 sequencing runs to date) that contamination with amplicons from previous runs will be picked up. The reviewer makes an excellent point though, that we should explicitly alert potential users of the sources of contamination, and suggest counteracting measures. We have prepared a separate table that goes through all these points (Suppl. Table 3) and refer to it in the text (line 504).

Regarding the reviewer's point about whether positive positions from previous runs come up as contaminants in subsequent runs: we analyzed two consecutive NovaSeq runs (though separated by a week). The first run used 192 i5/i7 combinations and the second run used a non-overlapping set of 192 i5/i7 combinations. We extracted 73 positions that had been positive in the first run and asked how many reads we observe in the second run, that carry those index combinations that were positive in the first run (see figure to the right, "Putative contamination"). We also extracted the number of reads for 43 index combinations that were never included in any of our runs and are thus likely the result of sequencing errors of the barcodes. The distribution of read number for both these sets are practically identical (1-5 reads), and dramatically different from 100 randomly selected positive samples from the second run (10^5 - 10^6 reads). We therefore think that the measures we take (and that we recommend to the reader in new Suppl. Table 3) can really minimize the risk of

contamination, and chose to not discuss this more extensively in the manuscript, but added the valuable suggestion of maintaining a logbook of positive positions into **Suppl. Table 3**. Thank you for pointing this out!

- Authors give a false impression that SARSeq is as sensitive as the gold-standard RT-qPCR whereas, in reality, SARSeq is as sensitive as RT-qPCR if there is no RNA-extraction done. However, gold-standard diagnostic RT-qPCR is always done with RNA extraction which allows for two orders of magnitude higher limit of detection (10 cp/ul vs. 0.1 cp/ul in this assay, compare <https://jcm.asm.org/content/58/9/e01535-20> tables 1 and 2 with Figure 3B in this paper). I think the authors should discuss this.

We thank you for pointing out this ambiguity, it seems we were not sufficiently clear about this important point. We ran two independent experiments to address sensitivity relative to the gold standard method.

- We compared SARSeq to RT-qPCR when using the same sample type i.e. crude QE boiled samples (**Fig. 4A-E**). Here we see, that SARSeq reaches exactly the same sensitivity. This is expected, because in essence it is the same method just coupled to a different readout. This section of the manuscript aims to highlight that the method as such is of comparable sensitivity.
- In **Fig. 4G, H** we compared SARSeq to the Gold Standard method performed in the State Agency AGES (Austrian FDA equivalent). These experiments are indeed comparing purified RNA followed by RT-qPCR versus crude extract and SARSeq. Here we indeed demonstrate, that SARSeq begins to miss cases beyond Ct ~36 that are still diagnosed by the Gold Standard method. This section therefore aims to highlight the differences between Sample preparation methods.

To avoid misunderstanding we have adapted the relevant text (i.e. **line 408-409, 426-428 and 438-440**) and hope it is now sufficiently clear. Thank you again for pointing out that this section may have resulted in a misunderstanding.

- How many hours have PCR machines run in order to do RT and PCR of 10,000 samples? How many machines were used? Running all those machines sounds like a logistical challenge and should be mentioned. How many hours and PCR machines would it take from gargle collection to declaration of positive samples for 10,000 samples? It would be useful to have estimates of the time that machines and people would need to process 10,000 samples from collection to end result (positive or negative sample) since time and machines could be prohibiting from testing so many samples.

The reviewer correctly points out that PCR machine access is the limiting factor of this method. To remove some of the burden on PCR machines and facilitate higher throughput, we tested whether the RT step can be performed in a humid incubator set to 55°C. This works reliably and we have added these data as a new **Suppl. Fig. 1E**. For the PCR step it's clear that proper thermocyclers are necessary. Nevertheless, a standard PCR machine can be purchased for ~3000 EUR and can in principle run 15 cycles in one day. So already for the first day of use the investment per sample is only 3000/(15*96), about 2EUR per sample and would be paid off. We are frequently running with 15-30 PCR machines in parallel currently. To help understand the logistics challenges of the approach we added a short section as you suggested, in the discussion (**lines 606-609**). To follow up on your example, 10,000 samples (100 Plates) could be run in our > 30 PCR machines in three cycles or one afternoon.

Minor comments:

- First sentence in the last paragraph on page 2 is not complete, please revise.

Oops! Thank you!

- "This pipeline also suppressed read misassignment across plates, such that the highly-positive positions of Fig. 3A, C, E were not detected on other plates run in the same experiment" this sentence references a wrong figure.

We apologize for the confusion. The sentence is meant to refer to **Suppl. Fig. 5B** (see text change on lines **306-308**), which shows that in the same run that contained the strongly positive samples shown in Fig. 3 A, C, E, equivalent positions on two completely negative plates, do not produce any reads. This shows that samples in other plates with multiple reads analyzed on the same flow cell do not result in false positives due to index hopping. We changed the text and figure legend of **Suppl. Fig. 5B** to make this point clearer.

- Please give a reference for this claim: "It is thought that infectious COVID-19 patients show viral titers of $>10^3/\mu\text{l}$ ". Is this number for gargles (of how many ml?) or swabs (of how many ml?).

This was estimated in Wölfel et al, Nature 2020, from measurements on swab samples collected in 3 ml of VTM. Samples with titers below 10^6 /ml never yielded viral isolates. We now specified the titer better and added the reference. Thank you.

- "9 out of 30 plates are shown in Fig. 3E", here Fig. 3F is meant.

Thank you for noticing this, it has been fixed.

Reviewer #2 (Remarks to the Author):

This manuscript by Yelangula et al. describes the development of SARSeq, an NGS-based SARS-CoV-2 diagnostic assay that allows to multiplex thousands of samples in one sequence run. SARSeq shows a lot of similarities with SwabSeq, but has added a few innovations, including testing of saliva samples, and an innovative method to barcode individual samples. In addition, SARSeq allows for the testing of multiple respiratory pathogens in one assay.

We agree with this overall assessment and thank for the overall positive summary. We would also thank for reviewing this manuscript during those days of crises.

This reviewer has a few points that need to be addressed:

- The manuscript is very technical, and it is sometimes difficult to follow, because minuscule details are highlighted that distract from the main message. The manuscript should also be shortened, and a lot of the details should go to the supplementary data.

We agree with the reviewer that the manuscript is very technical and this could risk a good flow of the storyline. However, for such a method, that is used to diagnose patients, we strongly feel that technical details to maintain a robust and reproducible pipeline matter and should thus be at the heart of this manuscript. Also, this enables further developments and adaptation. We hope the reviewer can appreciate our arguments.

- One of the strong points of SARSeq, the detection of multiple respiratory pathogens, requires more attention. First, the vast majority of the experiments were conducted in samples with only one pathogen. I'm worried that for samples that have both a low titer and high titer pathogen, the sensitivity of the low titer pathogen is affected. Secondly, the influenza and rhinovirus assays should be validated against a clinical qPCR assay (similar to the experiments done for SARS-CoV-2).

The reviewer is correct in that the majority of experiments were done on samples with individual pathogens. However, we did assess combinations of SARS-CoV-2 with other viruses in a range of concentrations (**Fig. 5**). We attempted to procure clinical samples with other viruses but there were no significant influenza viruses circulating in the summer and even now in fall/winter the numbers are quite low and although we contacted different specialists in Vienna we could not acquire such samples. This means that unfortunately, a meaningful validation against clinical assays is at the moment not really possible. We agree however that for clinical approval of SARSeq for monitoring all viruses such experiments are needed and we added this statement to the discussion (**line 625**).

- The authors should deposit the final protocol in a repository like protocols.io, github or something similar. Also, the SARSseq website you are referring to in the manuscript does not have any content.

We would ask the reviewer to refer to **Suppl. Table 1**, which contains a complete calculation matrix for all experimental steps. This, together with the detailed Materials and Methods section should be sufficient for anyone to implement SARSeq. Furthermore, we have shared all scripts for the analysis at <https://github.com/alex-stark-imp/SARSeq> and at <https://starklab.org>. Unfortunately, SARSeq.org is still not online indeed, but we mention that we will share future developments here and will do so. In fact, we plan a major add-on to SARSeq very soon in response to the new variants of SARS-CoV2, so we would prefer to maintain the reference.

Reviewer #1 (Remarks to the Author):

The authors have addressed my comments of review round #1

Reviewer #2 (Remarks to the Author):

Comments to the author:

- The manuscript is very technical, and it is sometimes difficult to follow, because minuscule details are highlighted that distract from the main message. The manuscript should also be shortened, and a lot of the details should go to the supplementary data.

We agree with the reviewer that the manuscript is very technical and this could risk a good flow of the storyline. However, for such a method, that is used to diagnose patients, we strongly feel that technical details to maintain a robust and reproducible pipeline matter and should thus be at the heart of this manuscript. Also, this enables further developments and adaptation. We hope the reviewer can appreciate our arguments.

The manuscript shows little to no changes to improve the readability compared to the previous version. I think the technology here presented is great, but the manuscript still suffers from a lack of clarity and flow.

- One of the strong points of SARSeq, the detection of multiple respiratory pathogens, requires more attention. First, the vast majority of the experiments were conducted in samples with only one pathogen. I'm worried that for samples that have both a low titer and high titer pathogen, the sensitivity of the low titer pathogen is affected. Secondly, the influenza and rhinovirus assays should be validated against a clinical qPCR assay (similar to the experiments done for SARS-CoV-2).

The reviewer is correct in that the majority of experiments were done on samples with individual pathogens. However, we did assess combinations of SARS-CoV-2 with other viruses in a range of concentrations (Fig. 5). We attempted to procure clinical samples with other viruses but there were no significant influenza viruses circulating in the summer and even now in fall/winter the numbers are quite low and although we contacted different specialists in Vienna we could not acquire such samples. This means that unfortunately, a meaningful validation against clinical assays is at the moment not really possible. We agree however that for clinical approval of SARSeq for monitoring all viruses such experiments are needed and we added this statement to the discussion (line 625).

It is very unfortunate that the authors were not able to obtain clinical samples with multiple respiratory pathogens. However, the authors could have done additional experiments using contrived samples with multiple pathogens. According to figure 5A, only 3 samples contained multiple pathogens, two of which had identical titers, and one only had a 2 log difference in titer. So not even all pathogen combinations were tested. It's difficult to draw any meaningful conclusions from a sample size that is so small. Also, the authors spike gargle with viral RNA and not with virus particles, and can thus not make the claim that "SARSeq can detect multiple respiratory viruses in a single reaction", which should become "SARSeq can detect RNA of multiple respiratory viruses in a single reaction"

In addition, the results for Influenza B need to be repeated. The authors clearly messed up the experiment (as described in L496-499). This happens every day in labs all over the world, but the correct thing to do is to repeat the experiment, because you can't conclude anything meaningful

from an experiment where your negative controls show up as positive. This needs to be addressed. A more fundamental point is that it is unclear how well respiratory viruses can be detected in gargle, and how this would affect the sensitivity of the assay vis-a-vis clinical qPCR. In the current form this paragraph is by far the weakest of all, and since the authors have not shown the willingness and effort to adapt it, this reviewer suggests removing it entirely from the manuscript.

- The authors should deposit the final protocol in a repository like protocols.io, github or something similar. Also, the SARSseq website you are referring to in the manuscript does not have any content.

We would ask the reviewer to refer to Suppl. Table 1, which contains a complete calculation matrix for all experimental steps. This, together with the detailed Materials and Methods section should be sufficient for anyone to implement SARSeq. Furthermore, we have shared all the scripts for the analysis at <https://github.com/alex-stark-imp/SARSeq> and at <https://starklab.org>. Unfortunately, SARSeq.org is still not online indeed, but we mention that we will share future developments here and will do so. In fact, we plan a major add-on to SARSeq very soon in response to the new variants of SARS-CoV2, so we would prefer to maintain the reference.

The website still has "Lorem ipsum..." as a placeholder for all content, which is very disappointing. The authors literally have had months to fix this and didn't. Since it is clearly not a priority for the authors, there are very little guarantees that this will ever be correctly implemented, and the link to the website needs to be removed from the manuscript.